# mPLUG-Owl3: Towards Long Image-Sequence Understanding in Multi-Modal Large Language Models

**Jiabo Ye**[1]    **Haiyang Xu**[1]    **Haowei Liu**    **Anwen Hu**    **Ming Yan**[2]
**Qi Qian**    **Ji Zhang**    **Fei Huang**    **Jingren Zhou**
Alibaba Group
{yejiabo.yjb, shuofeng.xhy, ym119608}@alibaba-inc.com

https://github.com/X-PLUG/mPLUG-Owl

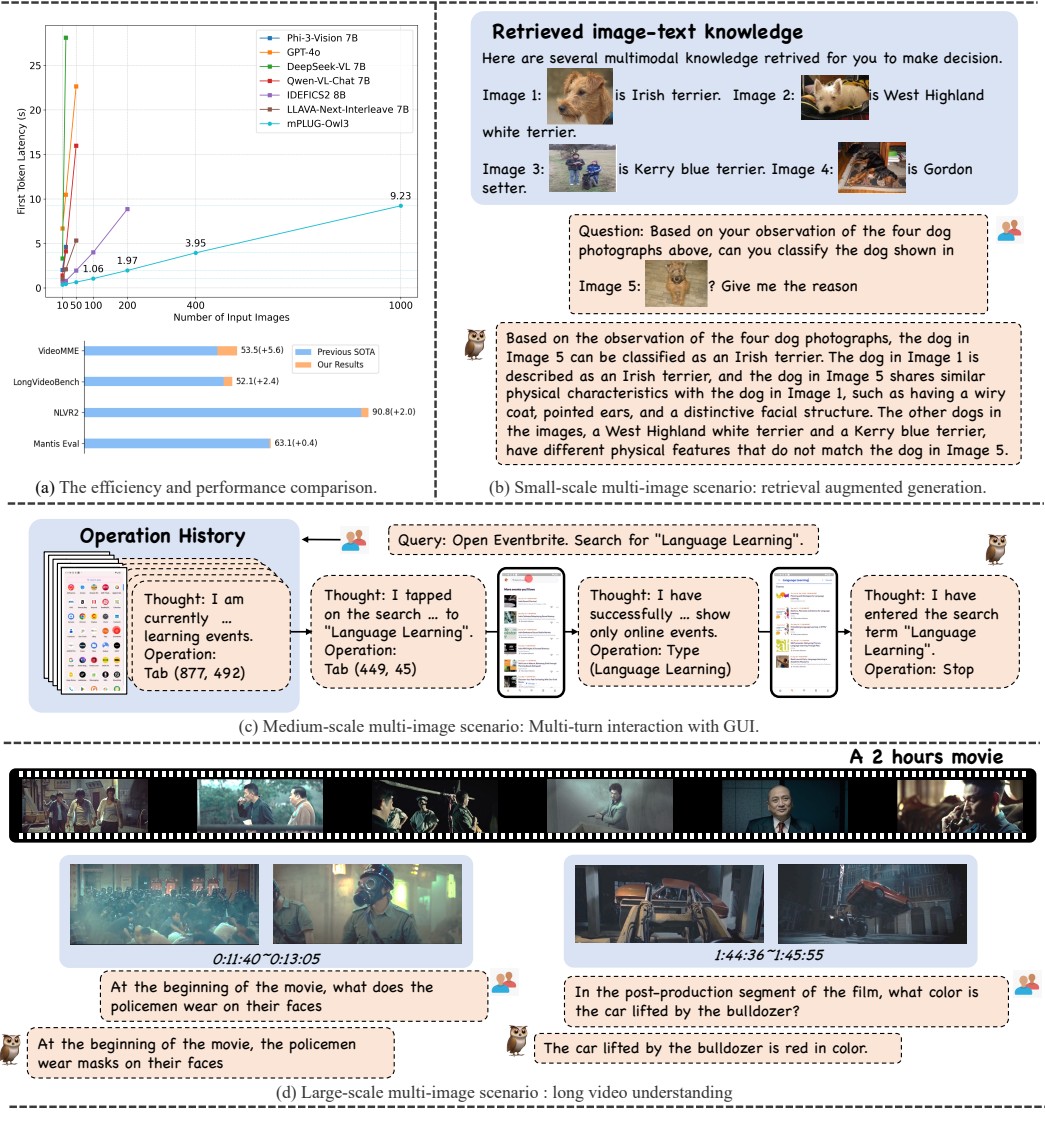

(a) The efficiency and performance comparison.

(b) Small-scale multi-image scenario: retrieval augmented generation.

(c) Medium-scale multi-image scenario: Multi-turn interaction with GUI.

(d) Large-scale multi-image scenario : long video understanding

Figure 1: (a) mPLUG-Owl3 demonstrates leading performance on video and multi-image understanding. (b,c,d) Examples of mPLUG-Owl3 on handling different scale of multi-image scenarios.

---

[1]Equal contribution

[2]Corresponding author

## ABSTRACT

Multi-modal Large Language Models have demonstrated remarkable capabilities in executing instructions for a variety of single-image tasks. Despite this progress, significant challenges remain in modeling long image sequences. In this work, we introduce the versatile multi-modal large language model, mPLUG-Owl3, which enhances the capability for long image-sequence understanding in scenarios that incorporate retrieved image-text knowledge, multimodal in-context examples, and lengthy videos. Specifically, we propose novel hyper attention blocks to efficiently integrate vision and language into a common language-guided semantic space, thereby facilitating the processing of extended multi-image scenarios. We conduct evaluations on 21 benchmarks that cover single/multi-image, and short/long video understanding. mPLUG-Owl3 achieves competitive performance with the state-of-the-art methods while reducing inference time and memory usage by 87.8% and 48.5% in average. Moreover, we propose a Distractor Resistance evaluation to assess the ability of models to maintain focus amidst distractions. mPLUG-Owl3 also demonstrates outstanding performance in distractor resistance on ultra-long visual sequence inputs. We hope that mPLUG-Owl3 can contribute to the development of more efficient and powerful multimodal large language models.

## 1 INTRODUCTION

Recently, Multimodal Large Languages Models (Liu et al., 2023a; Ye et al., 2023b; Liu et al., 2024a; Ye et al., 2024; Chen et al., 2024d) have achieved rapid advancements, demonstrating strong single-image understanding capabilities. The current approaches primarily rely on vast amounts of image and text data to align Large Language Models (Zheng et al., 2023; Touvron et al., 2023a;b) with visual encoders, thereby extending multimodal capabilities.

More advanced image-sequence understanding capabilities are required in practical applications, such as Multi-Image Reasoning (Suhr et al., 2018; Lu et al., 2021; Jiang et al., 2024), Multimodal RAG (Chen et al., 2022; Lin et al., 2024), Video Understanding (Xiao et al., 2021; Li et al., 2023c; Fu et al., 2024a; Wu et al., 2024), Multi-modal Agents (Wang et al., 2024; Zhang et al., 2024a), and Multi-Doc QA (Tito et al., 2023; Van Landeghem et al., 2023). The existing methods are primarily based on interleaved image-text web data for pre-training (Laurençon et al., 2023; Laurençon et al., 2024) to extend multi-image capabilities or focused on the in-context abilities (Alayrac et al., 2022; Awadalla et al., 2023; Zhao et al., 2023; Yang et al., 2023; Peng et al., 2024) within multi-image scenarios. However, these methods have not explored the in-depth comprehension or the efficiency of multi-images sufficiently, which makes it hard to support long image sequences. For example, LLAVA-Next-Interleave (Li et al., 2024) and IDEFICS2 (Laurençon et al., 2024) insert raw or compressed visual features into textual sequences. As shown in Figure 1 (a), the inference latency increases dramatically as the number of images increases. This design also costs more memory and hinders these models from processing more visual inputs.

To address this challenge, we introduce mPLUG-Owl3, a new general-purpose multi-modal foundation model. mPLUG-Owl3 is designed to handle long image sequences both effectively and efficiently. mPLUG-Owl3 integrates innovative hyper attention blocks in the language model to achieve efficient interleaved vision-language semantic alignment. Specifically, Hyper Attention introduces cross-attention parallel to the self-attention in the transformer block. The language query is reused to select and extract visual features from a lengthy visual sequence, allowing for adaptively obtaining complementary visual information that the language model lacks, based on textual semantics.

As shown in Figure 1 (b,c,d), the mPLUG-Owl3, characterized by its high inference efficiency, is also capable of handling a variety of tasks involving interleaved text and image modalities, including multimodal retrieval-augmented generation, multimodal in-context examples, and long video understanding. mPLUG-Owl3 also can handle inputs containing up to 1,000 images at an extremely fast speed surpassing the existing models. We evaluate mPLUG-Owl3 with a total of twenty-one benchmarks, spanning single-image, multi-image, short, and long video scenarios. Among models of the 8B-level size training on publicly available data, mPLUG-Owl3 achieves

state-of-the-art results in 15 out of 21 benchmarks. Besides existing benchmarks, we also propose a challenging long visual sequence evaluation named Distractor Resistance. It is designed to assess the ability of models to maintain focus amidst distractions. mPLUG-Owl3 demonstrates outstanding performance in handling ultra-long visual sequence inputs while also maintaining extremely high execution efficiency. It can also reduce inference time costs by 87.8% and memory usage by 48.5% in average compared to concatenate-based method. The superior performance of the new architecture in mPLUG-Owl3 implies a trend for future multimodal large language models.

## 2 MPLUG-OWL3

As illustrated in Figure 2, mPLUG-Owl3 comprises a visual encoder, a linear projection layer, and a decoder-only language model. This architecture is commonly employed in recently proposed Multi-modal Large Language Models. Unless specified otherwise, we use Siglip-400m (Zhai et al., 2023) as the visual encoder and Qwen2 (Yang et al., 2024) as the language model. First, we provide detailed information about our efficient architecture and its handling of various lengths of visual inputs in Section 2.1. Additionally, we introduce the Hyper Attention module in Section 2.2. It is a lightweight extension designed to enhance the transformer blocks of the language model by enabling cross-attention capabilities for adaptive visual sequence utilization.

### 2.1 CROSS-ATTENTION BASED ARCHITECTURE

Popular MLLMs (e.g., LLAVA-Interleave (Li et al., 2024), InternVL (Chen et al., 2024d)) insert visual features into the sequence of embeddings, which can easily exhaust the language model's context window, resulting in significant memory and computational overhead. This kind of disadvantage hinders these MLLMs to modeling the long vision input such as multiple images, videos and multiple pieces high-resolution images.

Therefore, mPLUG-Owl3 consider use cross-attention for feeding the visual information into the language model. Specifically, given a interleaved multimodal input $S = [T_1, I_1, T_2, I_2, T_3]$ (the format can be adapted to various text-image organizational structures), mPLUG-Owl3 first extract visual features of the input images and use a linear projection to align the dimensions of visual features to be the same of the language model. The projected visual features are denoted by $\mathbf{H_{img}} = [I_1^t, I_2^t] \in \mathbb{R}^{L \times D_t}$. The text sequence are $S_{text} = [T_1, T_{img}, T_2, T_{img}, T_3] \in \mathbb{R}^{L \times D_t}$, where $T_{img}$ is a plain text <|image|> to indicate the original place of the image. We feed the sequence into the word embedding to obtain text feature $\mathbf{H_{text}}$.

In the language model, we fuse the visual features $\mathbf{H_{img}}$ into the text features $\mathbf{H_{text}^i} \in \mathbb{R}^{L \times D_t}$ of the $i^{th}$ layer through cross-attention operator. Different from Flamingo (Alayrac et al., 2022) and EVLM (Chen et al., 2024b) that insert an additional layer into each layer of transformer layer, we sparsely extend a small number of transformer blocks in the network to perform cross attention parallel with self-attention. We name this the Hyper Attention Transformer Block (HATB). We discuss the design of HATB in detail in Section 2.2. HABT can significantly reduces the number of additional training parameters and facilitates model convergence. Besides, we observe that having fewer HATBs does not degrade the model's performance; instead, it offers the advantages of low memory consumption and high inference efficiency during inference. For a language model consisting of $N$ layers, we start from layer 0 and uniformly extend $K$ layers to HATB.

### 2.2 HYPER ATTENTION TRANSFORMER BLOCK

In this section, we specifically introduce the Hyper Attention Transformer Block used in mPLUG-Owl3. The cross-attention structure employed in Flamingo, as shown in Figure 3 (a), has been widely utilized in constructing MLLMs (e.g., IDEFICS (Laurençon et al., 2023), EVLM (Chen et al., 2024b)). However, this structure presents three main drawbacks: it introduces a large number of additional parameters, which results in significant memory and computational overhead; the knowledge learned by the language model cannot benefit the understanding of visual inputs; the cross attention does not fully take into account the original positions of images in the interleaved sequence, which limits the performance of these models in multi-image scenarios. In response to these issues, we propose a lightweight Hyper Attention Transformer Block, illustrated in Figure 3 (b). This block introduces a small number of parameters and extends self-attention capabilities

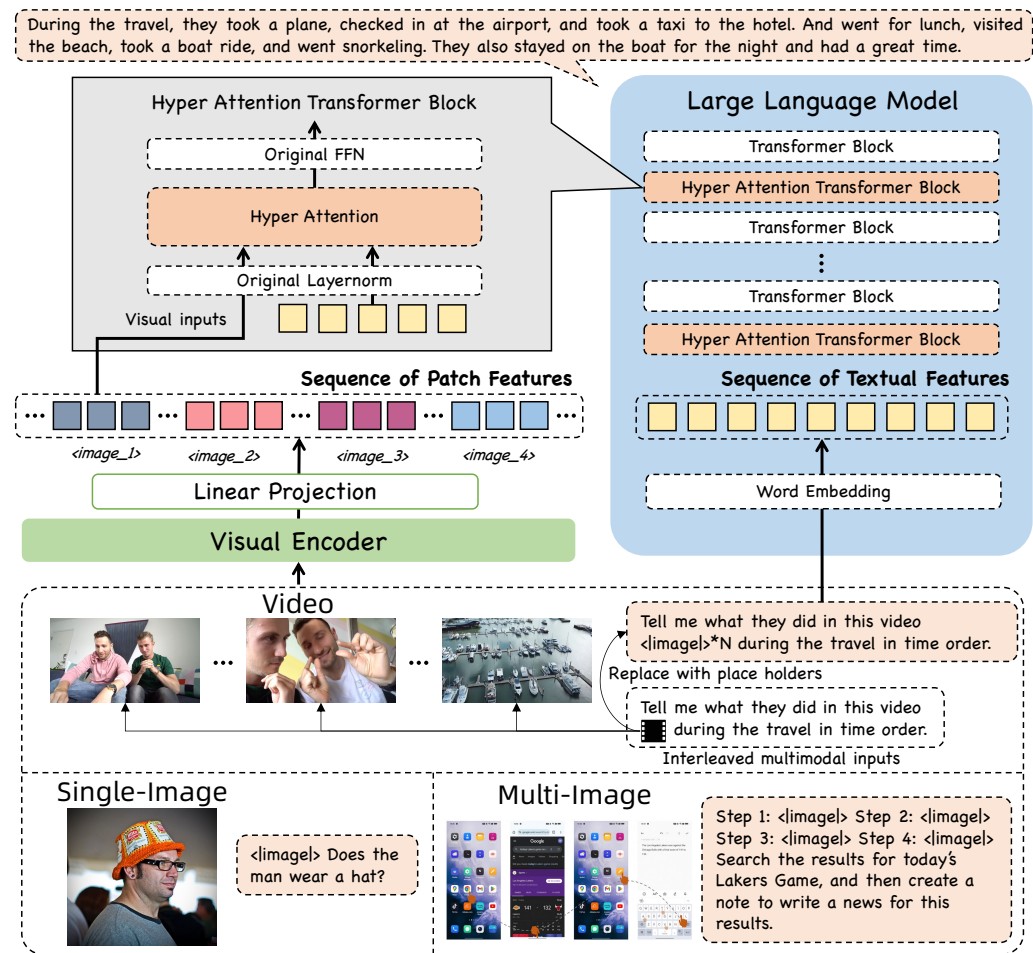

Figure 2: An overview of mPLUG-Owl3.

to perform both intra-text self-attention and inter-modal cross-attention between text and images in parallel. It also introduces a Multimodal-Interleaved Rotary Position Embedding (MI-Rope) to maintain the position information of images. The extended modifications are detailed below:

**Shared Input Layernorm.** The visual feature $\mathbf{H_{img}}$ and the $i^{th}$ layer's text features $\mathbf{H_{text}^i}$, although sharing the same dimensionality, originate from different distributions. Hence, both sets of features are initially normalized using a LayerNorm module. Our findings indicate that employing the LayerNorm module already integrated within the transformer block results in better convergence compared to training a separate layer normalization module specifically for the visual features.

**Modality-Specific Key-Value Projection.** In cross-attention, the **Query** is derived from textual data, while the **Key** and **Value** are extracted from visual features. Inspired by Ye et al. (2024), we construct a weight matrix $\mathbf{W}_{img}^{K\&V} \in \mathbb{R}^{2D \times D}$ to generate the **Key** and **Value** for the visual features. This matrix is initialized using the weights from the language model's KV (Key-Value) projection. Furthermore, the query vector from the self-attention mechanism is repurposed as the **Query** in the cross-attention. The computation procedure for self-attention remains unchanged. This design is beneficial as it preserves more specific visual information and allows for the adaptive supplementation of visual information that the language model lacks, based on textual semantics.

**Visual Position Modeling in Attention.** For models that process multiple images, positional encoding is essential to correctly understanding interleaved image-text input. Existing cross-attention models, such as Flamingo (Alayrac et al., 2022) and IDEFICS (Laurençon et al., 2023), do not assign

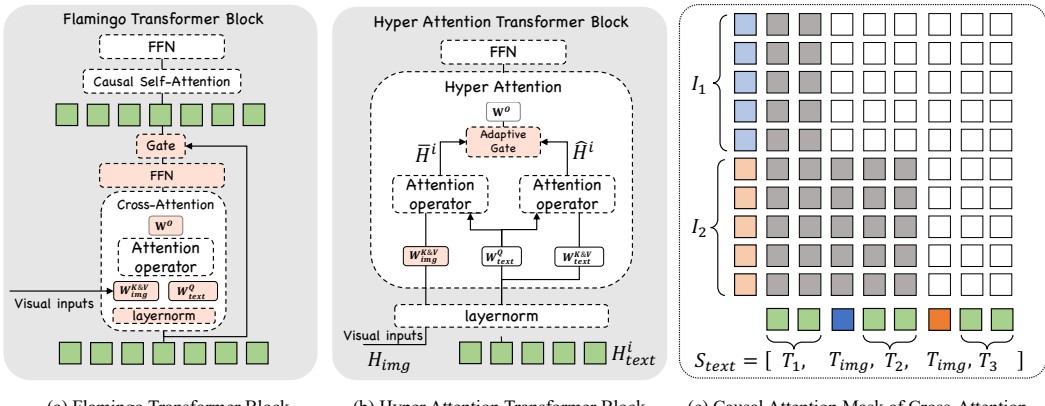

Figure 3: Comparison between Flamingo Transformer Block (a) and Hyper Attention Transformer Block (b). Pink indicates that the module is additionally introduced. (c) presents the causal attention mask strategy of cross attention in Hyper Attention in a image-text interleaved scenario. The gray block denotes the attention score is ignored. $T_{img}$ denotes the token of plain text <|image|>.

position embeddings to visual inputs, leading to suboptimal performance in scenarios involving multiple images. To accurately represent the original positions of images in interleaved sequences, we develop a Multimodal-Interleaved Rotary Position Embedding, which we name MI-Rope. Specifically, for each visual feature $I_n$ of image $n$, we pre-record the position index of its placeholder $T_{img}$ in the interleaved sequence $S_{text}$. All patches of $I_n$ share $T_n$'s positional encoding to obtain the rotary embedding. This ensures that the positional encoding of the image not only reflects the order among images but also reveals its position in the textual context. We also use a causal attention mask in cross attention. As shown in Figure 3 (c), for a text sequence $S = [T_1, T_{img}, T_2, T_{img}, T_3]$, each text token can only attend the visual features that precede it. Then, HATB simultaneously performs cross-attention and self-attention, denoting the resulting hidden states as $\bar{\mathbf{H}}^{\mathbf{i}}$ and $\hat{\mathbf{H}}^{\mathbf{i}}$.

**Adaptive Gating** Existing implementations of cross-attention utilize a learnable scale to regulate the extent of information transfer from the image to the language model. However, the semantics of language are ignored. Consequently, we introduce an adaptive gate that obtains the gate value based on the textual features. Gate value of each token is obtained by: $\mathbf{g} = \text{Sigmoid}(\mathbf{W}_{gate}^T \hat{\mathbf{H}}^{\mathbf{i}})$. Then, the multimodal features are fused: $\mathbf{H}_{\mathbf{fused}}^{\mathbf{i}} = \bar{\mathbf{H}}_{\mathbf{text}}^{\mathbf{i}} * \mathbf{g} + \hat{\mathbf{H}}_{\mathbf{text}}^{\mathbf{i}} * (1 - \mathbf{g})$. The $\mathbf{H}_{\mathbf{fused}}^{\mathbf{i}}$ is passed to the FFN and fed to the next layer of transformer.

# 3 IMPLEMENT DETAILS

## 3.1 TRAINING PARADIGM

We adopt a three-stage training approach for mPLUG-Owl3. Initially, we pre-train mPLUG-Owl3 using image-text pairs to achieve robust multimodal alignment. In the second stage, we leverage diverse datasets that include image and video captions to enhance the model's ability to understand multiple images. Finally, we fine-tune mPLUG-Owl3 using a mixture of supervised data, encompassing tasks involving both single and multiple images, to ensure comprehensive performance. The statistics of the datasets, the training settings and data processing details can be found in Appendix A. We train three sizes of mPLUG-Owl3, based on Qwen2 with sizes of 0.5B, 1.5B, and 7B. All three models share the same visual encoder.

### 3.1.1 PRE-TRAINING

We follow the mPLUG-Owl2 (Ye et al., 2024) to collect the pre-training datasets and randomly sample a subset consists of 41 million image-text pairs for pre-training. During pre-training, only the newly introduced modules are trainable, which include the linear layer following the vision encoder, the visual KV projection, and the Adaptive Gate in the Hyper Attention Transformer Block.

| Model | # Param | VQAv2 | OK-VQA | GQA | VizWizQA | TextVQA |
|---|---|---|---|---|---|---|
| CogVLM | 17B | 82.3 | 64.8 | - | - | 70.4 |
| EVLM-Chat | 32B | 81.9 | - | 64.4 | 47.3 | 67.5 |
| Flamingo | 80B | 81.3 | 50.6 | - | 57.2 | 54.7 |
| 8B-level MLLMs | | | | | | |
| Idefics1 | 9B | 68.8 | 50.4 | - | - | 39.3 |
| Flamingo | 9B | 51.8 | 44.7 | - | - | 46.3 |
| mPLUG-Owl2 | 8B | 79.4 | 57.7 | 56.1 | 54.5 | 58.2 |
| LLAVA-1.5 | 8B | 78.5 | - | 62.0 | 50.0 | 58.2 |
| LLAVA-Next | 8B | 81.8 | - | 64.2 | 57.6 | 64.9 |
| VILA-1.5 | 8B | 80.9 | - | 61.9 | 58.7 | 66.3 |
| Idefics2 | 8B | 80.8 | 53.5 | 51.1 | 58.5 | **70.4** |
| Mantis-SigLIP | 8B | 74.9 | 55.4 | 56.3 | 42.2 | 59.2 |
| mPLUG-Owl3 | 1B | 69.2 | 40.3 | 55.7 | 34.8 | 53.1 |
| | 2B | 79.3 | 53.2 | 61.0 | 56.6 | 62.6 |
| | 8B | **82.1** | **60.1** | **65.0** | **63.5** | 69.0 |

Table 1: **Performance comparison on visual question answering.** The accuracy is reported. We use **bold** to mark the highest score and underline to mark the second highest of 8B-level MLLMs.

### 3.1.2 MULTI-IMAGE PRE-TRAINING

In the multi-image pre-training stage, we collected three types of data to enhance the model's multi-image understanding capabilities: (1) Interleaved data. We utilize sources such as MMDU (Liu et al., 2024c) and M4-Instruction (Li et al., 2024) for multi-image data. Additionally, from LLaVA-Recap 558K, we randomly sample 3 to 6 images and combine their image-caption pairs into an interleaved format to create Interleaved Captions. We also consider sampling 4 images and requiring a description of one among them to form Selective Captions. (2) Text-rich data. We use text reading and key point generation data proposed by UReader (Ye et al., 2023a), enabling the model to reconstruct the text and extract key points contained within text-rich images. (3) Video data. We adopt annotated data from ShareGPTVideo (Zhang et al., 2024b), which includes 900K caption entries and 240K question-answering instances. We also incorporate Chinese and English video caption data from VATEX (Wang et al., 2019). During multi-image training, modules except visual transformer are trainable.

### 3.1.3 SUPERVISED-FINETUNING

In Supervised-Finetuning stage, mPLUG-Owl3 is trained with an extensive and diverse assembly of instruction tuning datasets aimed at enhancing its instruction-following capability. The datasets include LLaVA-SFT-665K (Liu et al., 2024a), The Cauldron (Laurençon et al., 2024), Mantis (Jiang et al., 2024), M4-Instruction (Li et al., 2024), ALLaVA (Chen et al., 2024a), ShareGPTVideo-QA 240K (Zhang et al., 2024b), Video Instruct 100K (Maaz et al., 2023), MSR-VTT (Xu et al., 2016) and MSVD Caption (Chen & Dolan, 2011). We keep the same training setting as the Multi-image Pre-training stage.

## 4 EXPERIMENTS

### 4.1 VISUAL QUESTION ANSWERING BENCHMARKS

We conduct experiments on VQAv2 (Goyal et al., 2016), OK-VQA (Marino et al., 2019), GQA (Hudson & Manning, 2019), VizWizQA (Bigham et al., 2010), and TextVQA (Singh et al., 2019). The detail introduction can be found in Appendix H.1. These datasets are strategically selected to thoroughly evaluate our model's ability to understand and reason across various visual contexts and knowledge domains.

Table 1 presents the comparison results between mPLUG-Owl3 and State-of-the-Art multimodal large language models trained on publicly available data. mPLUG-Owl3 outperforms 8B-level language models in VQAv2, OK-VQA, GQA, and VizWizQA. Furthermore, it surpasses the 32B-

| Model | # Param | MMB-EN | MMB-CN | MM-Vet | POPE | AI2D |
|-------|---------|--------|--------|--------|------|------|
| CogVLM | 17B | 65.8 | 69.8 | 52.8 | 88.0 | 63.3 |
| EVLM-Chat | 32B | 76.9 | 76.9 | - | 89.7 | 76.0 |
| *8B-level MLLMs* | | | | | | |
| LLAVA-1.5 | 8B | 64.3 | 58.3 | 31.1 | 85.9 | 55.5 |
| OpenFlamingo | 9B | 32.4 | 14.4 | 24.8 | - | 31.7 |
| mPLUG-Owl2 | 8B | 64.5 | - | 36.2 | - | 55.7 |
| LLAVA-Next | 8B | 67.4 | 60.6 | 43.9 | 86.5 | 66.6 |
| LLAVA-Interleave | 8B | - | - | 39.1 | 86.8 | 73.9 |
| VILA1.5 | 8B | 72.3 | 66.2 | 38.3 | 84.4 | - |
| Idefics2 | 8B | 75.7 | 68.6 | 34.0 | 86.2 | 72.3 |
| Cambrian | 8B | 74.6 | 67.9 | **48.1** | 86.4 | **74.6** |
| Mantis-SigLIP | 8B | 68.7 | 63.1 | 39.3 | 84.5 | 67.3 |
| | 1B | 51.4 | 44.8 | 21.4 | 85.3 | 46.7 |
| mPLUG-Owl3 | 2B | 65.7 | 61.8 | 36.3 | 87.4 | 62.6 |
| | 8B | **77.6** | **74.3** | 40.1 | **88.2** | 73.8 |

Table 2: **Zero-shot multi-modal evaluation on multi-modal benchmarks.** The overall scores are reported for evaluation. We use **bold** to mark the highest score and underline to mark the second highest of 8B-level MLLMs.

parameter EVLM in GQA and VizWizQA. In TextVQA, although mPLUG-Owl3's performance is slightly lower than that of Idefics2, it still exceeds that of other 8B models. It is noteworthy that, despite having 8B parameters, mPLUG-Owl3 exhibits superior inference speed and memory efficiency compared to models of the same scale, thanks to the introduction of Hyper Attention.

## 4.2 GENERAL MLLM BENCHMARKS

We evaluate mPLUG-Owl3 on various single-image general multimodal large language model benchmarks including MMBench (Liu et al., 2023b), MM-Vet (Yu et al., 2023), POPE (Li et al., 2023d) and AI2D (Kembhavi et al., 2016). Please refer to the Appendix H.1 for more details.

Table 2 shows that mPLUG-Owl3 achieves state-of-the-art performance on MMBench-EN, MMBench-CN, MM-Vet and POPE across 8B-level models. It also matches or surpasses the performance of larger models such as CogVLM (Wang et al., 2023) and EVLM-Chat (Chen et al., 2024b). mPLUG-Owl3 does not achieve state-of-the-art performance on the AI2D dataset. Due to limited training resources, we do not fine-tune the vision encoder, which restricts its performance in scenarios rich in text.

## 4.3 MULTI-IMAGE AND VIDEO BENCHMARK

We also evaluate the performance of mPLUG-Owl3 on video and multi-image benchmarks (including NextQA (Xiao et al., 2021), MVBench (Li et al., 2023c) VideoMME (Fu et al., 2024a), LongVideoBench (Wu et al., 2024) and MLVU (Zhou et al., 2024)), as it is capable of processing multiple images with an interleaved format. The introduction of benchmakrs and sample setting can be found in Appendix H.2.

The results of video evaluation is shown in Table 3. On benchmark that measure the short video understanding performance, mPLUG-Owl3 achieves performance comparable to state-of-the-art models. And on long video benchmarks, mPLUG-Owl3 significantly outperforms existing models and present a more significant advantage in understanding long videos. The results demonstrates that mPLUG-Owl3 is highly suitable for understanding videos with various durations. We also present qualitative results on video understanding in Appendix G.2.

Table 4 presents the the evaluation results on multi-image understanding. The evaluation benchmarks consist of NLVR2 (Suhr et al., 2018), Mantis-Eval (Jiang et al., 2024), MathVerse-mv (Li et al., 2024), SciVerse-mv (Li et al., 2024), BLINK (Fu et al., 2024b) and Q-Bench2 (Zhang et al., 2024c)). We provide a detail introduction in Appendix H.3.

| Model | # Param | NextQA | MVBench | VideoMME w/o sub | LongVideoBench-val | MLVU |
|---|---|---|---|---|---|---|
| VideoChat2 | 8B | 68.6 | 51.9 | 43.8 | 36.0 | 47.9 |
| Video-LLaMA2 | 8B | - | **54.6** | 47.9 | - | 48.5 |
| Video-ChatGPT | 8B | - | 32.7 | - | - | 31.3 |
| ShareGPT4Video | 8B | - | - | 39.9 | 39.7 | 46.4 |
| PLLaVA | 8B | - | 46.6 | - | 40.2 | - |
| Idefics2 | 8B | - | 29.7 | 25.9 | 49.7 | 49.8 |
| Mantis-SigLIP | 8B | - | 50.2 | 44.0 | 47.0 | 49.3 |
| LLAVA-Interleave | 8B | 78.2 | 53.1 | 48.7 | 48.8 | 56.4 |
| mPLUG-Owl3 | 1B | 65.3 | 47.1 | 41.8 | 42.8 | 52.7 |
| | 2B | 74.5 | 51.8 | 48.4 | 50.4 | 59.8 |
| | 8B | **78.6** | 54.5 | **53.5** | **52.1** | **63.7** |

Table 3: **Multi-modal evaluation on video understanding benchmarks.** The overall scores are reported for evaluation. We use **bold** to mark the highest score and underline to mark the second highest.

mPLUG-Owl3 surpasses existing models in both NLVR2 and Mantis-Eval. On MathVerse-mv and SciVerse-mv, it can be observed that mPLUG-Owl3 significantly outperforms LLaVA-Interleave. However, on BLINK, mPLUG-Owl3 performs weaker than LLaVA-Interleave. We note that this dataset requires models to possess low-level visual perception capabilities for fine details in images, and mPLUG-Owl3's ability may be limited due to the vision encoder being frozen during training. On the Q-Bench2, which evaluates a model's ability to discern low-level visual differences across multiple images globally, mPLUG-Owl3 achieves performance comparable to the state-of-the-art. We also conduct fine-grained multi-image experiments and present the result in Appendices B and C. mPLUG-Owl3 exhibits a stronger capability for following external multimodal knowledge. Additionally, we also present qualitative results of the multi-image understanding scenarios in Appendix G.1.

| **Model** | # Param | NLVR2 | Mantis-Eval | MathVerse-mv | SciVerse-mv | BLINK | Q-Bench2 |
|---|---|---|---|---|---|---|---|
| CogVLM | 17B | 58.6 | 45.2 | - | - | 41.5 | 53.2 |
| VideoLLaVA | 8B | 56.5 | 35.9 | - | - | 38.9 | 45.7 |
| VILA | 8B | 76.5 | 51.2 | - | - | 39.3 | 45.7 |
| Idefics2 | 8B | 86.9 | 48.9 | - | - | 45.2 | 57.0 |
| Mantis-SigLIP | 8B | 87.4 | 59.5 | - | - | 46.4 | 69.9 |
| LLAVA-Interleave | 8B | 88.8 | 62.7 | 32.8 | 31.6 | **52.6** | **74.2** |
| mPLUG-Owl3 | 1B | 75.8 | 39.6 | 23.9 | 26.4 | 40.6 | 50.6 |
| | 2B | 85.5 | 54.4 | 42.5 | 30.0 | 43.0 | 65.8 |
| | 8B | **90.8** | **63.1** | **65.0** | **86.2** | 50.3 | 74.0 |

Table 4: **Multi-modal evaluation on multi-image understanding benchmarks.** The overall scores are reported for evaluation. We use **bold** to mark the highest score and underline to mark the second highest.

## 4.4 ABLATION STUDIES

In this section, we investigate the integration method and design of Hyper Attention. We also validate the integration density and the results are presented in Appendix E. For ablation study experiments, we adopt the training methods of LLaVA-1.5 (Liu et al., 2024a) using the same datasets to conduct our ablation study. Additionally, we employ the Qwen1.5 7B as our language model. To validate the single-image understanding capabilities of our structures, we use datasets such as GQA and TextVQA (with OCR). We conducti zero-shot evaluations on benchmarks including MvBench, VideoMME, NLVR2, and Mantis-Eval to examine the generalization capabilities in multi-image understanding and video comprehension.

### 4.4.1 CROSS ATTENTION INTEGRATION

There are two primary methods to integrate Cross-Attention into the transformer block: one method positions it prior to the self-attention (referred to as Pre-Cross-Attention), while the other places it

| Attention Structure | Elapsed Time (ms) | VRAM (GB) | GQA | TextVQA | MvBench | VideoMME | NLVR2 | Mantis-Eval |
|---|---|---|---|---|---|---|---|---|
| Concatenate | 2761.7 | 40.6 | **59.0** | **51.6** | 22.4 | 25.1 | 55.7 | 38.7 |
| Pre-Cross-Attention | 384.3 | 21.2 | 53.8 | 45.2 | **43.0** | 38.9 | 55.3 | 44.7 |
| Post-Cross-Attention | 397.2 | 21.2 | 48.9 | 40.9 | 38.3 | 37.0 | 54.0 | 47.0 |
| Hyper Attention | **336.9** | **20.9** | 57.6 | 50.0 | 42.8 | **39.4** | **59.5** | **51.6** |

Table 5: Comparison between different attention structure. Concatenate means direct concatenate visual and text feature sequences. We use **bold** to mark the highest score.

| Adaptive Gating | Shared LayerNorm | MI-Rope | GQA | TextVQA | MvBench | VideoMME | NLVR2 | Mantis |
|---|---|---|---|---|---|---|---|---|
| | | | 53.3 | 44.6 | 40.2 | 38.1 | 52.7 | 41.9 |
| ✓ | | | 55.7 | 49.3 | **43.2** | **40.1** | 53.4 | 47.9 |
| ✓ | ✓ | | **58.1** | 49.7 | 42.8 | 38.4 | 54.9 | 46.1 |
| ✓ | ✓ | ✓ | 57.6 | **50.0** | 42.8 | 39.4 | **59.5** | **51.6** |

Table 6: Ablation on the Adaptive Gating, Shared LayerNorm and MI-Rope.

subsequent to the self-attention (referred to as Post-Cross-Attention). We analyze both configurations and compare them to the concatenate-based method and our novel Hyper Attention in mPLUG-Owl3.

Table 5 shows that the concatenate-based model which directly embeds image features into the input sequence of the language model, has the best performance in single-image understanding. On the other hand, utilizing Post-Cross-Attention results in the worst performance. Comparatively, Pre-Cross-Attention performs better but still incurs some performance loss. Hyper Attention, however, achieves comparable performance with concatenate-based model. In evaluations involving videos and multiple images, we observe that the concatenate-based model demonstrate poor performance. This is attributed to the inadequate training of inter-image attention, which significantly disrupts the model's hidden states. Conversely, both single images and multiple images share the same paradigm when performing cross attention with text, which allows its multi-image capability to be better generalized from single-image training. the Hyper Attention design stands out as particularly effective in balancing the model's capabilities for handling both single and multiple images, showcasing superior generalizability. It also reduces inference time costs by 87.8% and memory usage by 48.5% in average compared to concatenate-based method.

### 4.4.2 DESIGN OF HYPER ATTENTION

To further investigate the impact of the structural design of Hyper Attention on model performance, we start with a basic hyper attention model and gradually introduce adaptive gating, shared layer-norm, and MI-Rope. The Table 6 shows that when incorporate adaptive gating, the single-image understanding performance is significantly improved. And if we use a shared layernorm, performance is further improved. In video scenario, we notice that even without any inter-image position encoding, the performance of video understanding is also improved, suggesting the temporality inherent in visual content can also be implicitly modeled by the model with the help of adaptive gating. However, when evaluating models with multiple images, the contextual position of the images is crucial and cannot be implicitly modeled. Therefore, it can be observed that incorporating adaptive gating and shared layernorm does not lead to performance improvement on multi-image benchmarks. However, with the introduction of MI-Rope, the metrics for various multi-image benchmarks have demonstrated significant improvement.

### 4.5 DISTRACTOR RESISTANCE IN LONG VISUAL CONTEXTS

We develop a challenge evaluation method to assess the distractor resistance of multimodal models in long visual contexts. Specifically, based on MMBench, we randomly insert $N$ Distractor images, where $N = 1, 5, 10, 20, 50, 100, 200, 400$ and measure the model's accuracy. The details of the experiment setting and analysis can be found in Appendix D.

In Figure 4, LLaVA and Mantis struggles to consistently answer the questions accurately when 50 distractors images are present, resulting in a low accuracy rate. In contrast, mPLUG-Owl3 only drops to a performance level of 43.09%. As the images increases to 400, the performance of mPLUG-Owl3 decreases to 28.58%. It is still a challenging task and valuable for future models to evaluate.

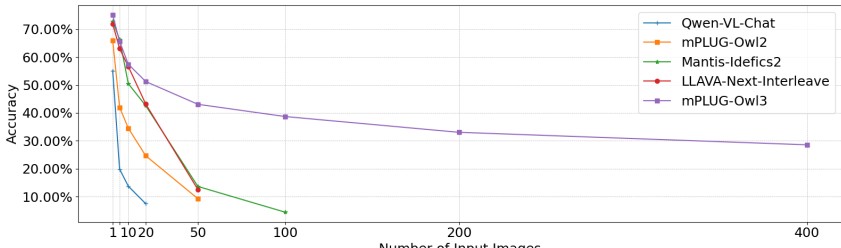

Figure 4: The performance of interference resistance with long visual context across LLaVA-Next-Interleave 7B, Mantis-Idefics2, Qwen-VL, mPLUG-Owl2 and mPLUG-Owl3.

## 5 RELATED WORK

### 5.1 MULTIMODAL LARGE LANGUAGE MODELS

Multimodal large language models (MLLMs) can perceive visual contents, conduct visual reasoning, and engage in multimodal dialogue with humans.

Models like LLaVA (Liu et al., 2023a) use an MLP to map visual features into the representation space of the language model, and directly concatenate them with the text sequence. These methods can preserve fine-grained visual information but consume a large number of tokens which slows down both training and inference. To reduce the number of tokens, models such as mPLUG-Owl (Ye et al., 2023b) and IDEFICS2 (Laurençon et al., 2024) adopt a structure similar to Q-Former (Li et al., 2023a), compressing the token count to a fixed size. Flamingo (Alayrac et al., 2022) first proposed embedding cross-attention layers into the language model, integrating visual features into the intermediate representations of the language model. IDEFICS (Laurençon et al., 2023) and EVLM (Chen et al., 2024b) have also trained MLLMs based on this structure. This method avoids occupying the context window of the LLM. However, it introduces more parameters and may interfere with the intermediate representations of the pre-trained language models, making the performance of such models often sub-optimal compared to mainstream models.

### 5.2 MULTIMODAL MODELS WITH INTERLEAVED SUPPORT

Recent research are expanding the capabilities of multimodal models to process multiple images inputs.

Video is a special form of multi-image existence, and MLLMs related to video understanding treat frames as multiple images with temporal correlation as input. These models such as VideoChat2 (Li et al., 2023b) and VideoLLaMA2 (Cheng et al., 2024) introduce extra module to compress temporal frames. ShareGPT4Video (Chen et al., 2024c) proposes to improve the video understanding by introducing GPT-4 annotated video caption as pretrain data. In general multi-image scenario, including in-context learning, cross image reference, comparison, and reasoning. Flamingo (Alayrac et al., 2022) demonstrates limited in-context learning capabilities, while Idefics2 (Laurençon et al., 2024) and LLAVA-Interleave (Li et al., 2024) further enhance the model's multi-image understanding capabilities by constructing more refined multi-image training data.

## 6 CONCLUSION

In this paper, we present mPLUG-Owl3, a multi-modal large language model that significantly advances the state-of-the-art in handling both single-image, multi-image and video tasks. The introduction of novel Hyper Attention enables the mPLUG-Owl3 to maintain the fine-grained visual input and effectively fuse visual and textual information, leading to superior performance across various benchmarks. We also propose a challenging long visual sequence evaluation named Distractor Resistance. Notably, mPLUG-Owl3 excels in managing ultra-long visual sequences and demonstrates a strong performance in evaluation. We believe that mPLUG-Owl3 reveals a direction for building efficient and effective multi-modal large language models.

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

# A  TRAINING DETAILS

| Stage 1: Pretraining | | Stage 2: Multi-Image Training | | Stage 3: Self-Supervised Fintuning | |
|---|---|---|---|---|---|
| Dataset Name | Percentage | Dataset Name | Percentage | Dataset Name | Percentage |
| DataComp-1B | 35.22% | ShareGPTVideo | 34.63% | LLAVA-SFT | 57.95% |
| LAION-en | 26.07% | Selective Caption | 19.29% | The Cauldron | 12.50% |
| COYO-700M | 14.47% | M4-Instruction | 16.69% | Mantis | 10.41% |
| COYO-700M-OCR | 9.60% | VATEX | 15.77% | M4-Instruction | 9.26% |
| LAION-zh | 7.73% | Text Reading | 7.36% | ALLAVA | 6.95% |
| Wukong | 5.64% | Interleaved Caption | 5.25% | ShareGPTVideo-QA | 2.02% |
| CC12M | 0.81% | MMDU | 1.01% | Video Instruct | 0.84% |
| Others | 0.46% | - | - | MSRTT/MSVD Caption | 0.06% |

Table 7: Dataset percentages used in Pretraining, Multi-Image Training, and Self-Supervised Fintuning. Others include CC3M, OCR-CC, COCO and SBU.

| Setting | | | Stage 1: Pretraining | Stage 2: Multi-Image Training | Stage 3: Self-Supervised Fintuning |
|---|---|---|---|---|---|
| Training | Learning Rate (Max, Min) | | (1e-3, 1e-5) | (2e-5, 1e-7) | (2e-5, 1e-7) |
| | Global Batch Size | | 2048 | 1024 | 1024 |
| | Training Steps | | 20K | 3K | 11K |
| | Warmup ratio | | 0.03 | | |
| | Trainable Modules | | Linear Projection Visual KV Projection Adaptive Gate | Linear Projection Full Language Model | Linear Projection Full Language Model |
| Model | Resolution | | $384^2$ | up to $384^2 \times 6$ | up to $384^2 \times 6$ |
| | Sequence Length | | 768 | 4096 | 4096 |
| Accelerating | Precision | | Mixed-precision FP16/BF16 | | |
| | ZeRO Optimization | | Zero-1 | | |
| | Gradient Checkpointing | | No. | Yes. | Yes. |
| | Model Parallel | | TP=1 | TP=4 | TP=4 |

Table 8: The training settings across three stages: Pretraining, Multi-Image Training, and Self-Supervised Finetuning.

## A.1  TRAINING DATA PROCESSING

For high-resolution images, inspired by UReader (Ye et al., 2023a), we introduce a similar adaptive method for image cropping. For a given image, we select from the cropping grids (2,2), (1,3), (1,4), (3,1), (4,1), (2,3), and (3,2) that most closely matches the shape of the input image. Additionally, we retain a global version of the original image. During the Supervised-Finetuning stage, for datasets rich in text, we perform cropping with a probability of 100%. For datasets containing a single image without text, we apply cropping with a probability of 20%. For datasets containing multiple images or videos, we do not perform cropping. During evaluation, cropping is enabled only for single-image tasks.

For videos, we sample 8 frames per video by default. Meanwhile, we replace the video markers in the text with multiple <|image|> placeholders corresponding to the number of sampled frames.

## B  FINE-GRAINED INVESTIGATION OF CAPABILITIES IN MULTI-IMAGE SCENARIOS

To more comprehensively investigate the fine-grained abilities of mPLUG-Owl3 in multi-image scenarios, we conduct experiments on MI-Bench (Liu et al., 2024b), a recently proposed multi-image benchmark. We exclude Fine-Grained Visual Recognition from evaluation because it consists of images from mini-ImageNet that may have been seen by existing models.

Table 9 shown that mPLUG-Owl3 achieves state-of-the-art performance on aspects of General Comparison, Subtle Difference, Temporal Reasoning, Logical Reasoning and Text-Rich Images across popular open-sourced MLLMs. It also outperform GPT-4V and GPT-4o on General Comparison.

| Model | GC | SD | VR | TR | LR | TRI | VTK | TVK |
|---|---|---|---|---|---|---|---|---|
| Closed-source MLLMs | | | | | | | | |
| GPT-4o | 80.7 | 90.5 | 46.8 | 68.0 | 69.8 | 74.8 | 54.7 | 63.3 |
| GPT-4V | 72.8 | 79.2 | 45.8 | 61.8 | 66.3 | 71.0 | 52.0 | 56.0 |
| Open-source MLLMs | | | | | | | | |
| mPLUG-Owl2 | 64.2 | 40.1 | 35.6 | 30.7 | 41.3 | 39.0 | 17.0 | 25.6 |
| MMICL | 53.7 | 46.4 | **41.1** | **47.0** | 59.6 | 27.6 | 22.1 | 35.9 |
| Idefics2-I | 83.1 | 49.7 | 32.6 | 44.8 | 56.4 | 43.9 | 25.6 | 39.0 |
| Mantis | 83.0 | 54.1 | 37.6 | 45.5 | 63.4 | 37.7 | 26.4 | 41.7 |
| mPLUG-Owl3 | **86.4** | **70.1** | 33.0 | 46.8 | **67.2** | **50.1** | **34.0** | **48.8** |

Table 9: **Multi-image evaluation on MI-Bench (Liu et al., 2024b)**. We use **bold** to mark the highest score of open-sourced multimodel large language models. The evaluation consists of the following tasks: General Comparison (GC), Subtle Difference (SD), Visual Referring (VR), Temporal Reasoning (TR), Logical Reasoning (LR), Text-Rich Images (TRI), Vision-linked Textual Knowledge (VTK) and Text-linked Visual Knowledge (TVK).

The results demonstrates that our model possesses robust capabilities in various multi-image input scenarios. The Hyper Attention structure of mPLUG-Owl3 better preserves the original visual features, enabling it to excel in single-image tasks as well. And this type of multimodal knowledge also assists it in more accurately completing multi-image tasks.

## C  VISUAL-LINK KNOWLEDGE ENHANCE

Based on MI-Bench, we further examine the performance in scenarios involving supplementary visual knowledge and present the results in Table 10. Specifically, we test the models' performance in the Vision-linked Textual Knowledge task under conditions with and without additional visual knowledge. Since the provided visual knowledge does not always correspond to the images related to the questions, this setup more accurately simulates the performance of models in real-world Retrieval-Augmented Generation systems. Results show that LLaVA-Interleave achieves higher accuracy without external knowledge, while noisy interleaved text and image inputs severely degrade its performance. In contrast, mPLUG-Owl3 is capable of identifying relevant visual knowledge from noisy inputs, thereby enhancing its accuracy.

| Models | w/o Visual-link Knowledge | w/ Visual-link Knowledge |
|---|---|---|
| LLaVA-Interleave | 31.2 | 28.7 (-2.5) |
| mPLUG-Owl3 | 32.1 | 34.0 (+1.9) |

Table 10: Visual-link Knowledge Enhance experiment between LLaVA-Interleave and mPLUG-Owl3.

## D  DISTRACTOR RESISTANCE IN LONG VISUAL CONTEXTS

We notice that multimodal models, when modeling multiple images, are susceptible to interference from surrounding images, leading to visual illusions. Therefore, we develop a challenge evaluation method to assess the distractor resistance of multimodal models in long visual contexts.

Specifically, we take samples from the MMBench dev set. For each test sample, we randomly select $N-1$ images from the original MMBench dev set as distractor and construct the model input in the format of *Image 1: <|image|> Image 2: <|image|> ... Image N: <|image|>. In Image X, {question}*, where $N = 1, 5, 10, 20, 50, 100, 200, 400$ and $X$ denotes the index of the image corresponding to the question. We use the CircularEval to measure the accuracy scores. For each question, we construct test samples with different orders of options and varying distractor images. The model needs to

| Hyper Attention Layers Indices | GQA | TextVQA | MvBench | VideoMME | NLVR2 | Mantis-Eval |
|---|---|---|---|---|---|---|
| [9, 27] | 55.1 | **51.3** | 42.2 | 38.2 | 58.3 | 48.4 |
| [1, 5, 9, 13, 17, 21, 25, 29] | 56.2 | 48.3 | 41.5 | **39.5** | 52.4 | 47.5 |
| [1, 9, 17, 25] | **57.6** | 50.0 | **42.8** | 39.4 | **59.5** | **51.6** |

Table 11: Comparison between different layers for integrating hyper attention structures. We use **bold** to mark the highest score.

answer all test samples for a given question correctly for it to be counted as correct. Consequently, as the number of distractor images increases, the evaluation becomes significantly more challenging.

The results are shown in Figure 4. It can be observed that the introduction of distractor images results in a certain degree of performance loss for all the models. When the number of images reaches 20 and 50, the performance of LLaVA-Next-Interleave dramatically drops to 43.18% and 12.52%, respectively. We observe that when the number of images reaches 50, LLaVA struggles to consistently answer the questions accurately when different distractor images are present, resulting in a low accuracy rate. And when the number of images reaches 100, Mantis-Idefics2 fails to solve most of the problems correctly. In contrast, mPLUG-Owl3 only drops to a performance level of 43.09% when processing 50 images. As the number of images increases to 400, the performance of mPLUG-Owl3 decreases to 28.58%. Since our multi-image training data consists of only about 6-8 images, this also presents a challenge for our model. Nonetheless, mPLUG-Owl3 can serve as a baseline for future research.

# E    IMPACT OF INTEGRATION DENSITY OF HYPER ATTENTION

We explore the integration density of the hyper attention. As shown in Table 11 the results indicate that even with just two layers of Hyper Attention, the model achieves impressive performance on single-image benchmarks, while also demonstrating generalization capabilities for videos and multiple images. However, when we apply a denser integration strategy by introducing eight layers of Hyper Attention, we find that it does not yield improved single-image performance at this scale of training data, and its zero-shot generalization is even worse. Therefore, we ultimately integrate only four layers into the entire model.

# F    PERFORMANCE OF SMALL-SIZED MODELS

Small-scale language models can be more easily deployed on the edge, and mPLUG-Owl3 can further enhance the efficiency of these smaller language models on multimodal scenario. We provide the performance of mPLUG-Owl3 based on 0.5B and 1.5B language models across single-image, video, and multi-image understanding tasks in Table 1, Table 2, Table 3 and Table 4. The results indicate that for single-image understanding, the 1B and 2B models exhibit an average performance drop of 19.1% and 6.7% compared to the 8B model. Meanwhile, we notice that the 2B-sized model achieving performance comparable to LLaVA-Next 8B. In the video domain, we find that the smaller models do not suffer much performance loss (averaging 10.5% and 3.5%); even the mPLUG-Owl3 2B outperforms the 8B scale model on long video understanding tasks. However, in multi-image understanding tasks, we observe a more significant performance drop for the smaller models, with the 1B and 2B models showing declines of 28.8% and 18%, respectively. This is because understanding relationships among multiple images is more reliant on the capabilities of the language model.

# G    QUALITATIVE RESULTS

mPLUG-Owl3 can handle various number of images and videos as inputs. In this section, we further investigate the ability of mPLUG-Owl3 in real-world dialogue scenarios.

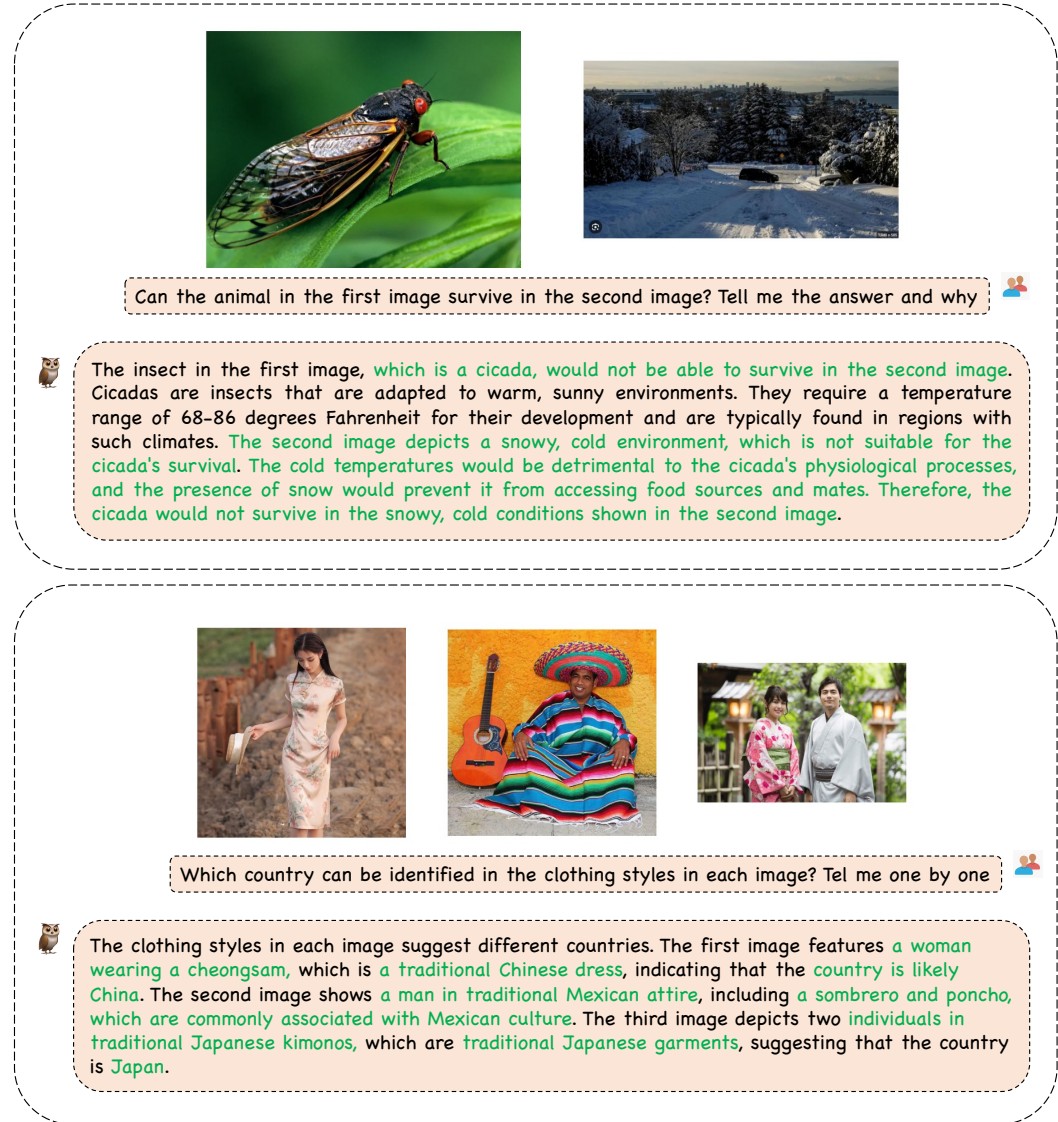

Figure 5: Examples for Multi-Image Understanding. We highlight the correct answers in green.

### G.1 Multi-Image Understanding

mPLUG-Owl3 demonstrate state-of-the-art performance on multi-image understanding benchmarks. In this section, we present multi-image dialogue examples in real-world. In the first example shown in Figure 5, it can be observed that mPLUG-Owl3 can activate the knowledge it learned based on the content of the images and perform cross-image reasoning. The second example demonstrates that the model can accurately distinguish the content of multiple images and respond appropriately based on cultural knowledge.

Figure 6 shows a multi-turn dialogue example. mPLUG-Owl3 can find the differences between two images in various views. Besides, it can describe the correlations between images.

As shown in Figure 7, mPLUG-Owl3 can maintain continuous interaction with a smartphone. During this process, the model effectively understands the multimodal interaction history and makes accurate decisions using a chain-of-thought approach. Additionally, it retains the initial user instructions without any degradation throughout the interaction.

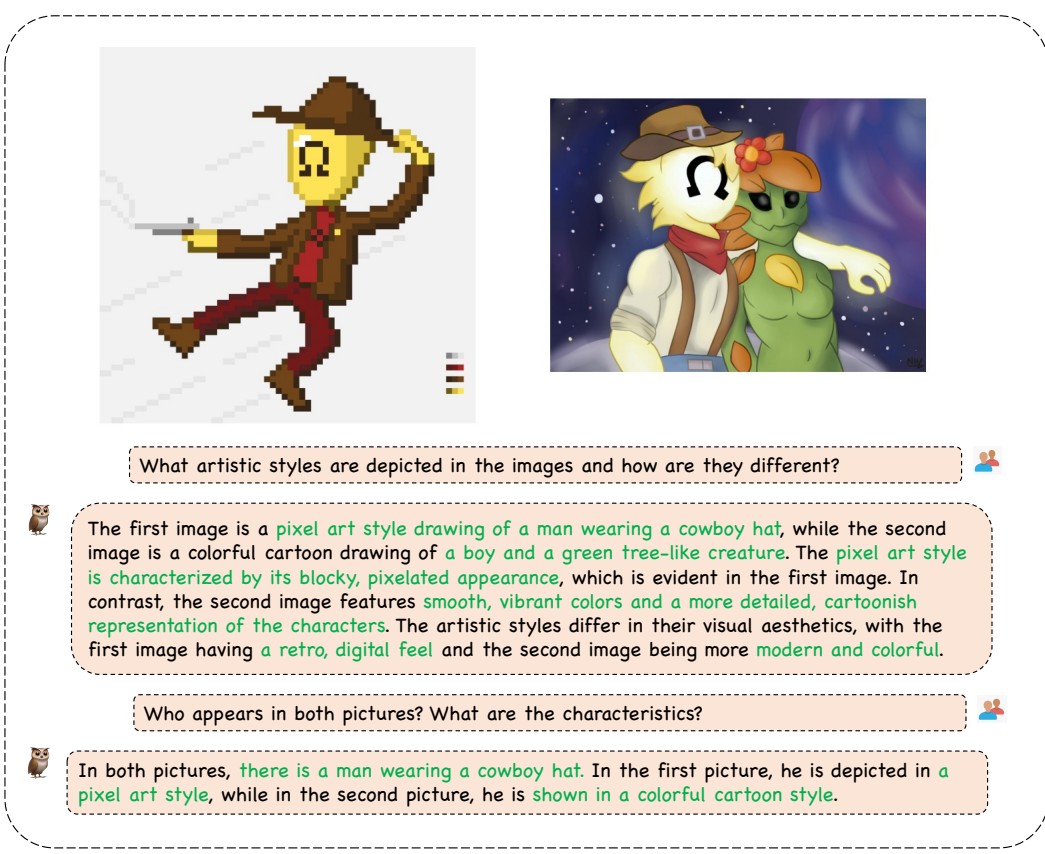

Figure 6: Examples for Multi-turn Multi-Image Dialogue. We highlight the correct answers in green.

## G.2 VIDEO UNDERSTANDING

We showcase the video understanding capabilities of mPLUG-Owl3. First, we compare it with LLaVA-Next-Interleave in Short Video Question Answering, Long Video Fine-grained Question Answering, and Long Video Comprehensive Understanding. For LLaVA-Next-Interleave, we input 8 frames, while for mPLUG-Owl3, we input 128 frames, which are the maximum numbers of images that can be accommodated by the two models on a V100-32G. The samples are shown in Figure 8.

In the short video tests, both LLaVA and mPLUG-Owl3 can provide correct answers. mPLUG-Owl3 tends to describe the attributes of objects based on the actual content seen. In long video lasting more than 40 minutes, when we ask about a specific detail, LLaVA fails to handle the long sequence and loses fine-grained information, rendering it unable to provide accurate information. On the other hand, mPLUG-Owl3 accurately captures key segment information within a long video. Additional, we have both models summarize the content of a longer video. mPLUG-Owl3's response is very detailed, not only providing an overall summary but also introducing the process in order. LLaVA-Next-Interleave's response, however, is more general and lacks detail. The comparative results indicate that mPLUG-Owl3 not only efficiently encodes long visual sequences but also captures and effectively utilizes both global and local information.

We also test mPLUG-Owl3 in multiple rounds using a long video that featuring many scenes. For clarity, we place the relevant segments beside the dialogue in the figure. During the test, we input only the complete video to the model. The dialogue is shown in Figure 9.

First, we ask a question with a temporal constraint, and mPLUG-Owl3 accurately understands the concept of "at first" and correctly describes the detail of "sitting in a room and discussing something on their laptops." However, the response incorrectly counts the number of people. The segment has only two people. We find that the model is confused by a later scene involving more people. We

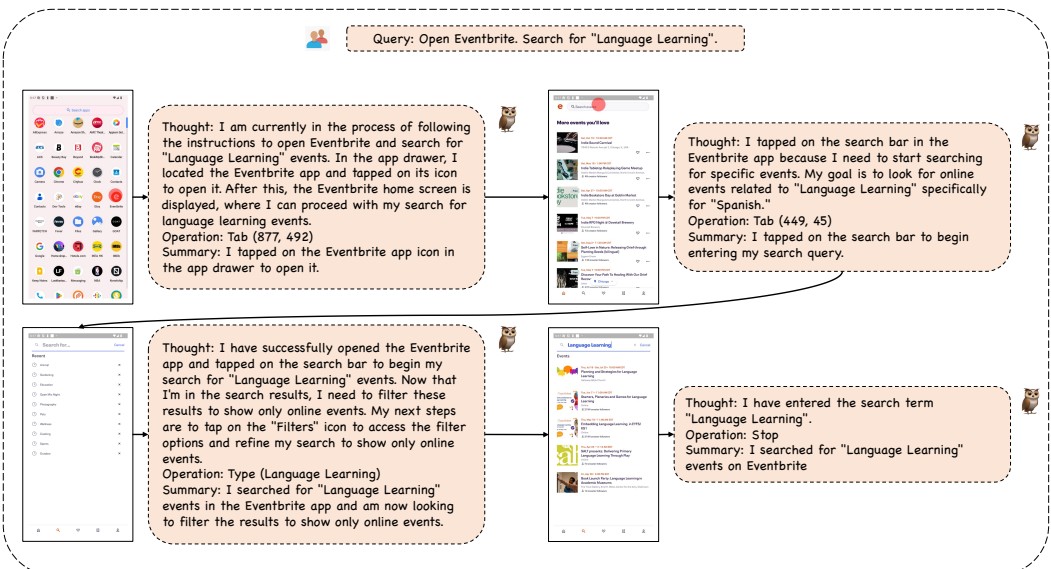

Figure 7: Example of multi-image understanding interactions between mPLUG-Owl3 and the smart-phone GUI in multi-round interactions. The tap operation is annotated by a red sphere.

also notice that the visual content of this segment does not involve Australia as a destination, but the model can infer this from some diagrams later in the video, which makes the response more detailed. Then, we ask about the camera brand in a frame that briefly appears, and mPLUG-Owl3 accurately notices the "Canon" logo in the image and provides the correct answer. Finally, we ask the model to describe the travel in order of time. We use the same color to identify the content described by the model and the corresponding video segments. Since the video involves many scenes and events, this poses a great challenge to the model. It can be observed that mPLUG-Owl3 accurately details the travel according to the timeline of the video. However, we also notice that mPLUG-Owl3 exhibits some hallucinations, incorrectly interpreting the reefs captured in the video as a beach. Additionally, while the activities on the boat happen during the day, mPLUG-Owl3, influenced by other nighttime scenes, makes an incorrect statement.

## H EVALUATION BENCHMARKS

### H.1 SINGLE-IMAGE BENCHMAKRS

We conduct experiments on a diverse set of visual question answering benchmarks, including VQAv2 (Goyal et al., 2016), OK-VQA (Marino et al., 2019), GQA (Hudson & Manning, 2019), VizWizQA (Bigham et al., 2010), and TextVQA (Singh et al., 2019). The VQAv2 dataset is currently the largest visual question answering dataset available. OK-VQA involves questions that require external knowledge beyond multimodal inputs. GQA is designed to validate the model's reasoning capabilities. VizWizQA is constructed from question-answer pairs sourced from visually impaired users. TextVQA focuses more on evaluating the model's ability to understand text in natural scenes.

We also evaluate mPLUG-Owl3 on various single-image general multimodal large language model benchmarks. MMBench (Liu et al., 2023b) provides a comprehensive evaluation of a model's multimodal capabilities in both Chinese and English contexts. MM-Vet (Yu et al., 2023) assesses the multimodal conversational abilities of a model using GPT-4 evaluation. POPE (Li et al., 2023d) can evaluate the extent of multimodal hallucinations in a model. AI2D (Kembhavi et al., 2016) assesses a model's ability to understand science diagrams inputs.

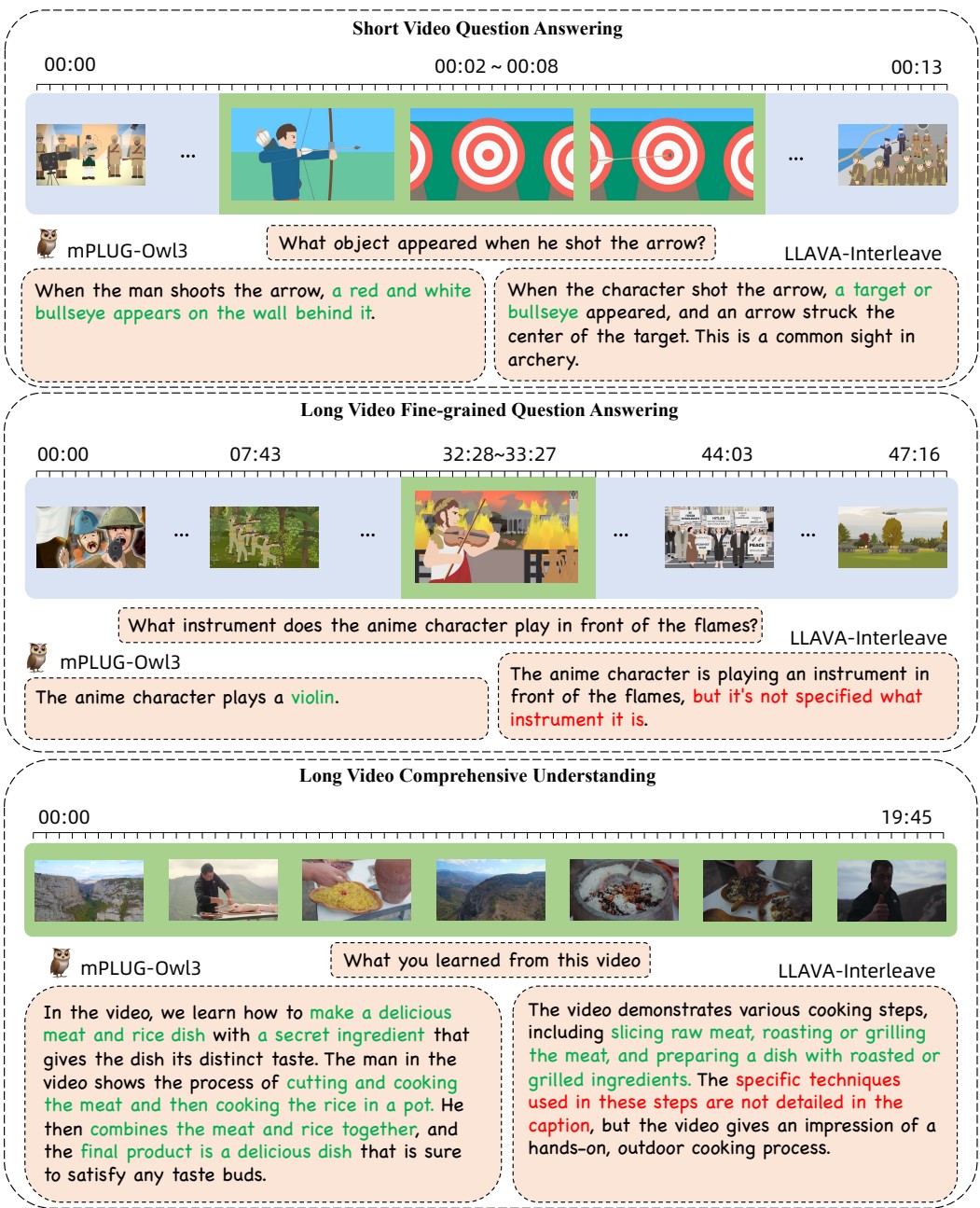

Figure 8: Comparison between mPLUG-Owl3 and LLaVA-Interleave across Short Video Question Answering, Long Video Fine-grained Question Answering, and Long Video Comprehensive Understanding. We highlight the correct and relevant parts of the answers in green, while the parts that fail to answer the question correctly are marked in red. Additionally, the segments of the video that are relevant to the questions are highlighted with a green background.

## H.2 VIDEO BENCHMARKS

We conduct experiments on a diverse set of video understanding benchmarks, including Next-tQA (Xiao et al., 2021) and MVBench (Li et al., 2023c), which are short video benchmarks with video durations all less than one minute. Thus, we sample 8 frames and 16 frames, separately. For benchmarks like VideoMME (Fu et al., 2024a), LongVideoBench (Wu et al., 2024), and MLVU (Zhou et al., 2024), which involve longer video durations up to one hour. For , we

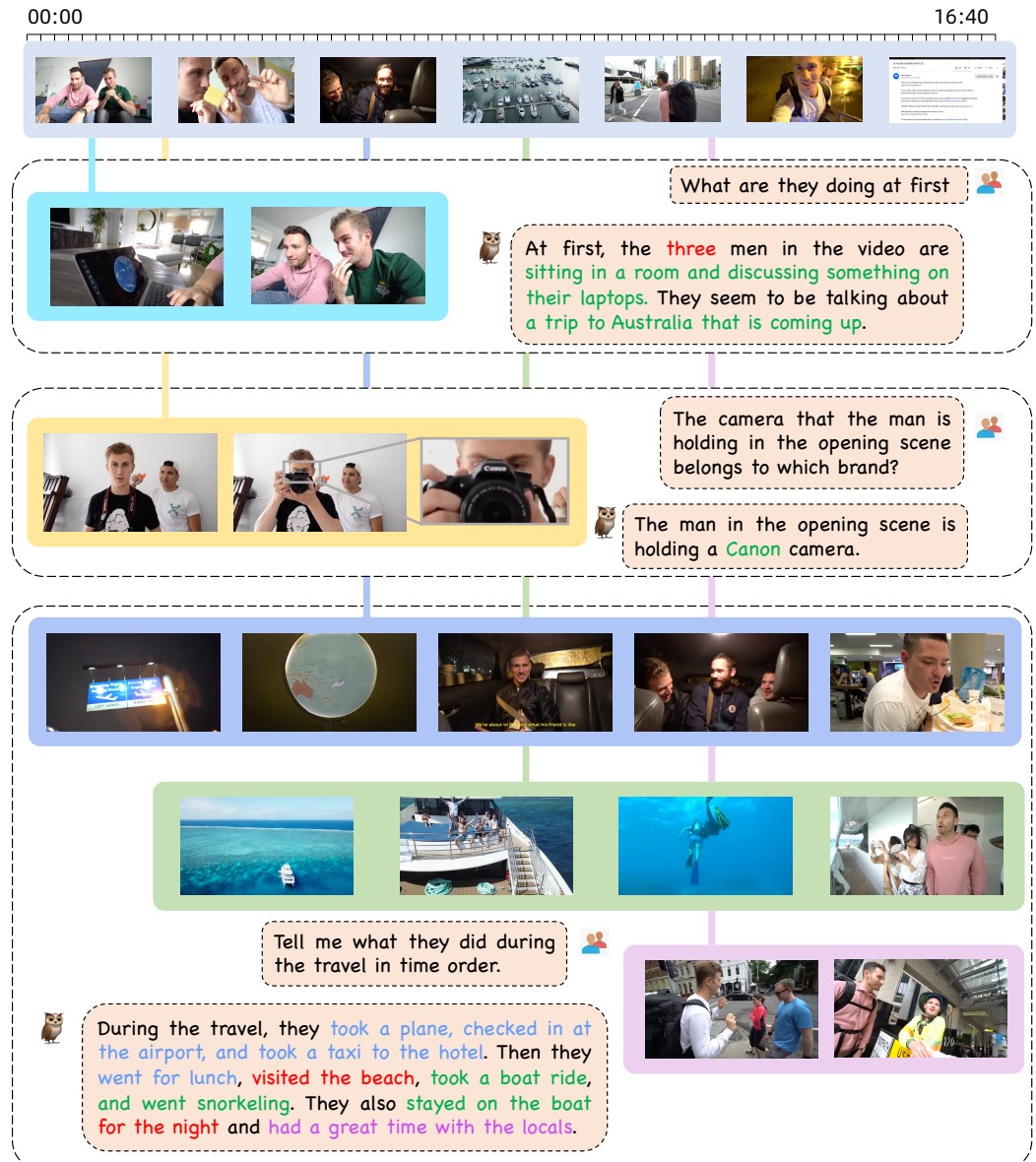

Figure 9: Examples of mPLUG-Owl3's understanding of complex video content

### H.3 MULTI-IMAGES BENCHMARKS

We conduct experiments on a diverse set of multi-image reasoning benchmarks, including NLVR2 (Suhr et al., 2018), Mantis-Eval (Jiang et al., 2024), MathVerse-mv (Li et al., 2024), SciVerse-mv (Li et al., 2024), BLINK (Fu et al., 2024b), and Q-Bench2 (Zhang et al., 2024c). NLVR2 tests the model's ability to perform logical reasoning based on the content of multiple images. Mantis-Eval evaluates the model's reasoning skills in complex scenarios involving multiple images. MathVerse-mv and SciVerse-mv assess the model's multi-image mathematical and scientific capabilities. We use the version released by llava-next-interleave for comparison with its reported results. BLINK and Q-Bench2 test the model's multi-image ability based on low-level visual perception.

