# OpenReview forum: "mPLUG-Owl3: Towards Long Image-Sequence Understanding in Multi-Modal Large Language Models"
_ICLR.cc/2025/Conference — ICLR 2025 Poster_

### Official Review · Reviewer_KHjw · 2024-11-02

**Soundness:** 3
**Presentation:** 3
**Contribution:** 3
**Rating:** 8
**Confidence:** 4

**Summary:**

This paper presents a novel multi-modal large language model, mPLUG-Owl3, designed for effective and efficient multi-image understanding. The core design is a novel hyper attention block that integrates visual and textual information. Comprehensive evaluations across 21 benchmarks—including single/multi-image and short/long video understanding—demonstrate the strong performance of mPLUG-Owl3. Additionally, the authors introduce a Distractor Resistance evaluation to assess the model's ability to maintain focus in the presence of distractions.

**Strengths:**

1.	The hyper attention block is an applaudable design that effectively fuses visual and textual information while preserving fine-grained visual input.
2.	The detailed ablations in Section 4.4 convincingly demonstrate the effectiveness of key design choices in the proposed model.
3.	The model shows strong performance across a wide range of image and video benchmarks.

**Weaknesses:**

1.	Although the proposed model demonstrates strong performance across a broad range of benchmarks, the comparison with other models lacks fairness from a data perspective. It would be beneficial to discuss the influence of data mixing and to establish at least one setting that is directly comparable to existing models.
2.	What is the impact of joint training on single-image, multi-image, and video data?
3.	Why does mPLUG-Owl3 freeze the vision encoder, considering that this design choice may adversely affect performance on BLINK, as discussed in Section 4.3?

**Questions:**

1.	What is the training cost associated with mPLUG-Owl3?
2.	How does mPLUG-Owl3 perform in terms of OCR, given that the hyper attention block could also be beneficial for high-resolution images?

---

> ### Author Response · Authors · 2024-11-22
> **KHjw Q1: Comparison on exact the same setting**
>
> Since mainstream models use private data and do not disclose training details, making fair comparisons with them is very difficult. Consequently, such comparisons do not yield meaningful insights. Given the limited time for rebuttal and the lower training cost of mPLUG-Owl3 compared to mainstream MLLMs, we are considering re-training mPLUG-Owl3 using a fair training setup. To ensure a fair and rigorous comparison, we will adopt exactly the same training settings and data as LLaVA-OneVision to train mPLUG-Owl3. This will allow us to compare mPLUG-Owl3 with LLaVA-OneVision, a strong baseline that uses publicly available data.
>
>
>
> The experimental results indicate that:
>
> + mPLUG-Owl3 and LLaVA-OneVision have comparable performance in single image scenarios.
> + When the models are faced with multi-image and video tasks, mPLUG-Owl3 outperforms LLaVA-OneVision by 2.7% and 2.8% in average, respectively.
>
> Meanwhile, as shown in Figure 1 (a) and Table 5, mPLUG-Owl3 also demonstrates significant advantages in memory consumption and inference time.
>
> |   | VQAv2  |  GQA | VizWizQA  |  MMB-EN  | POPE  |
> | --- | --- | --- | --- | --- | --- |
> | LLaVA-OneVison | **84.3** | 62.3 | 61.0 | **80.8** | **88.6** |
> | mPLUG-Owl3 | 83.2 | **64.7** | **62.9** | 80.4 | 88.1 |
>
>
> |   |  NLVR2  | Mantis-Eval  | MathVerse-mv  | SciVerse-mv  | BLINK  | Q-Bench2 |
> | --- | --- | --- | --- | --- | --- | --- |
> | LLaVA-OneVision 7B | 89.4 | 64.2 | **67.6** | 79.1 | 48.2 | 74.5 |
> | mPLUG-Owl3 7B | **92.7** | **67.3** | 65.1 | **82.7** | **53.8** | **77.7** |
>
>
> |   |  NextQA  | MVBench  | VideoMME w/o sub  | LongVideoBench-val  | MLVU  |
> | --- | --- | --- | --- | --- | --- |
> | LLaVA-OneVision 7B | 79.4 | 56.7 | 57.4 | 58.8 | 64.7 |
> | mPLUG-Owl3 7B | **82.3** | **59.5** | **59.3** | **59.8** | **70.0** |

---

> ### Author Response · Authors · 2024-11-22
> **KHjw Q2: Impact of joint training on single-image, multi-image, and video data.**
>
> We aim to develop a unified model capable of handling single-image, multi-image, and video inputs. Consequently, we are considering a joint training approach, which is also used by mainstream methods (e.g., IDEFICS2 [1] and LLaVA-Next-Interleaved [2]).
>
>
>
> To investigate the impact of joint training, we present the results of mPLUG-Owl3, which was first trained on single-image data, and then on multi-image and video data.
>
>
>
> The results indicate that training on multiple images and videos can cause the model to forget basic visual understanding abilities, which are helpful for solving complex multi-image and video tasks. As such, simply training on single-image data first and then on multi-image video data results in the model being sub-optimal in various scenarios.
>
> |   |  GQA | TextVQA | MvBench | VideoMME | NLVR2 | Mantis-Eval |
> | --- | --- | --- | --- | --- | --- | --- |
> | mPLUG-Owl3 | **65.0** | **69.0** | **54.5** | **53.5** | **90.8** | **63.1** |
> | mPLUG-Owl3 (SIMI) | 63.2 | 67.9 | 54.4 | 52.8 | 90.0 | 61.7 |
>
>
> Due to the significant resource overhead of exploring data mixing, we are considering assigning more resources to optimizing and validating the design of Hyper Attention. The training strategy is not the primary contribution of this paper, as we mainly followed insights from previous work.
>
>
>
> [1] Laurençon, Hugo, et al. "What matters when building vision-language models?." arXiv preprint arXiv:2405.02246 (2024).
>
> [2] Li, Feng, et al. "Llava-next-interleave: Tackling multi-image, video, and 3d in large multimodal models." arXiv preprint arXiv:2407.07895 (2024).

---

> ### Author Response · Authors · 2024-11-22
> **KHjw Q3: Why does mPLUG-Owl3 freeze the vision encoder?**
>
> We found that training the visual encoder reduces the training speed to one-third of its original speed. Due to limited training resources, we considered freezing the visual encoder in previous experiments.
>
> In the additional experiment addressing KHjw Q1, we strictly adhered to the setup of LLaVA OneVision and unfroze the visual encoder during the SFT phase. Compared to LLaVA OneVision, mPLUG-Owl3 achieves a significant performance advantage in BLINK (53.8 vs. 48.2). In conclusion, freezing the visual encoder does indeed affect the performance on BLINK, and unfreezing it can resolve the issue.

---

> ### Author Response · Authors · 2024-11-22
> **KHjw Q4: What is the training cost associated with mPLUG-Owl3?**
>
> For mPLUG-Owl3 7B, in stage 1 we used 32 V100 GPUs to train for about 4 days. In stages 1.5 and 2, we used 64 A100 GPUs to train for 12 hours and 25 hours, respectively.

---

> ### Author Response · Authors · 2024-11-22
> **KHjw Q5: How does mPLUG-Owl3 perform in terms of OCR？**
>
> This proposal is very constructive. Recent mainstream methods [1,2] adopt a splitting strategy for high-resolution images. Hyper Attention allows the mPLUG-Owl3 to split high-resolution images into more sub-images to retain more details. To investigate its impact, we use OCRBench for zero-shot evaluation to explore the performance of mPLUG-Owl3 in various splitting settings.
>
> ## **Experiment setup**
> We compare with the baselines Including Mantis-Siglip, VILA 1.5 and LLaVA-Next-Interleave.
>
> The splitting strategy divides a high-resolution image into a maximum of NUM_SUB_IMG sub-images, with each sub-image having a resolution of H×W that is compatible with the model's visual encoder. Consequently, the input resolution for the model can be expressed as H×W×NUM_SUB_IMG.
>
> + Mantis-Siglip does not use the splitting strategy, so NUM_SUB_IMG=1.
> + VILA 1.5 and LLaVA-Next-Interleave can split an image into a maximum of 4 images, so NUM_SUB_IMG=4.
> + We apply NUM_SUB_IMG = [4, 6, 12] for mPLUG-Owl3.
>
> ## **Results and analysis**
> The results are presented below. It can be observed that:
>
> + When using the same splitting strategy as LLaVA-Next-Interleave, which splits into up to 4 images, mPLUG-Owl3 achieves comparable performance to LLaVA-Next-Interleave and VILA 1.5. However, Mantis-Siglip significantly compresses the resolution of the input image, resulting in a lower score.
> + mPLUG-Owl3 can gain an advantage in OCR capabilities by splitting the input into up to 6 images, resulting in a resolution of 384×384×6.
> + Additionally, we increase the NUM_SUB_IMG to 12. Specifically, the candidate splitting grids for mPLUG-Owl3 are set to (2,2), (2,6), (2,4), (6,2), (4,1), (2,3), (3,2), (3,4), and (4,3), allowing mPLUG-Owl3 to handle a maximum resolution of 384×384×12. Under this setting, its performance on OCRBench improves further.
>
> | Models | Param | Resolution (H×W×NUM_SUB_IMG) | OCRBench |
> | --- | --- | --- | --- |
> | Mantis-Siglip | 8B | 384×384×1 | 347 |
> | VILA 1.5 | 8B | 384×384×4 | 438 |
> | LLaVA-Next-Interleave | 8B | 384×384×4 | 460 |
> | mPLUG-Owl3 | 8B | 384×384×4 | 476 |
> | mPLUG-Owl3 | 8B | 384×384×6 | 492 |
> | mPLUG-Owl3 | 8B | 384×384×12 | 548 |
>
>
> [1] Li, Feng, et al. "Llava-next-interleave: Tackling multi-image, video, and 3d in large multimodal models." arXiv preprint arXiv:2407.07895 (2024).
>
> [2] Lin, Ji, et al. "Vila: On pre-training for visual language models." Proceedings of the IEEE/CVF Conference on Computer Vision and Pattern Recognition. 2024.

---

> > ### Comment · Reviewer_KHjw · 2024-11-25
> >
> > Thank you to the authors for their significant efforts during the rebuttal. My main concerns have been addressed, and I am now leaning toward accepting this submission.

---

> > > ### Author Response · Authors · 2024-11-25
> > >
> > > Thank you for your volunteer review. Your suggestions and insightful comments play important roles in the rebuttal process to enhance the quality of our work. If any of your concerns have been addressed, could you please consider increasing the score?

---

> > > > ### Comment · Reviewer_KHjw · 2024-11-27
> > > >
> > > > Sure, I have increased my score to 8.

---

> > > > > ### Author Response · Authors · 2024-11-27
> > > > >
> > > > > Thank you again for your valuable comments and recognition of our paper!

---

### Official Review · Reviewer_qTJ2 · 2024-11-02

**Soundness:** 3
**Presentation:** 3
**Contribution:** 3
**Rating:** 8
**Confidence:** 3

**Summary:**

mPLUG-Owl3 is a new multi-modal foundation model that efficiently processes long image sequences using hyper attention blocks. The model can handle up to 1,000 images at high speeds and excels in various tasks including retrieval-augmented generation and video understanding. Among 8B-level models, it achieves state-of-the-art results in 15 out of 21 benchmarks while reducing inference time by 87.8% and memory usage by 48.5% compared to traditional methods.

**Strengths:**

The paper provides comprehensive coverage of existing benchmarks across various domains (including math reasoning, (NLVR), Mantis evaluations, and others), effectively comparing their hyper attention module against existing methods.

The authors present clarity in explaining the Hyper Attention module and its distinct differences from existing methods, particularly highlighting its efficiency improvements and architectural advantages.

The research addresses the important and challenging problem of long image sequence understanding in MLLMs, which is important as current models primarily focus on single-image tasks while real-world applications often require processing multiple images or video frames.

**Weaknesses:**

The paper compares mPLUG-Owl3 against diverse architectures (Mantis-SigLip, Cambrian, Idefics2) while acknowledging their different pretraining approaches, but doesn't isolate the benefits of Hyper Attention from other variables like training data.

**Questions:**

How do the effects of your pretraining and fine-tuning data look without the hyper attention module? Can we attest the performance gain to the hyper attention module? Just want a fair standard of comparison between models you are comparing against since you have different pertaining and fine-tuning methods compared to previous models in addition to the hyper attention module.

---

> ### Author Response · Authors · 2024-11-22
> **qTJ2 Q1: Attest the performance gain to the hyper attention module.**
>
> ## **The effects of hyper attention discussed in the current manuscript**
> In Table 5 of the manuscript, we compared the performance differences between Concatenate (which can be regarded as mPLUG-Owl3 without Hyper Attention) and Hyper Attention under the same data and training settings.
>
> The results suggest that model with Hyper Attention has comparable single image performance and better generalization across multiple images. Meanwhile, Hyper Attention can significantly reduce inference time and memory consumption.
>
> |  | Elapsed Time (ms) | GPU Memory (GB) | GQA | TextVQA | MvBench | VideoMME | NLVR2 | Mantis-Eval |
> | --- | --- | --- | --- | --- | --- | --- | --- | --- |
> | w/o Hyper Attention | 2761.7 | 40.6 | **59.0** | **51.6** | 22.4 | 25.1 | 55.7 | 38.7 |
> | w/ Hyper Attention | **336.9** | **20.9** | 57.6 | 50.0 | **42.8** | **39.4** | **59.5** | **51.6** |
>
>
> ## **Attest of the hyper attention on LLaVA-OneVision training data**
> In response to the qTJ2 Q2, we conduct model training and result comparison under the LLaVA-OneVision setup. It can also be regarded as an ablation experiment for Hyper Attention, because both mPLUG-Owl3 and LLaVA-OneVision use the same language model and visual encoder.

---

> ### Author Response · Authors · 2024-11-22
> **qTJ2 Q2:  A fair standard of comparison between baselines**
>
> Since mainstream models are trained on different datasets and training setups, making fair comparisons with them is very difficult. Given the limited time for rebuttal and the lower training cost of mPLUG-Owl3 compared to mainstream MLLMs, we are considering re-training mPLUG-Owl3 using a fair training setup.
>
> To ensure a fair and rigorous comparison, we will adopt exactly the same training settings and data as LLaVA-OneVision to train mPLUG-Owl3. This will allow us to compare mPLUG-Owl3 with LLaVA-OneVision, a strong baseline that uses publicly available data.
>
>
>
> The experimental results indicate that:
>
> + mPLUG-Owl3 and LLaVA-OneVision have comparable performance in single image scenarios.
> + When the models are faced with multi-image and video tasks, mPLUG-Owl3 outperforms LLaVA-OneVision by 2.7% and 2.8% in average, respectively.
>
> Meanwhile, as shown in Figure 1 (a) and Table 5, mPLUG-Owl3 also demonstrates significant advantages in memory consumption and inference time.
>
> |   | VQAv2  |  GQA | VizWizQA  |  MMB-EN  | POPE  |
> | --- | --- | --- | --- | --- | --- |
> | LLaVA-OneVison | **84.3** | 62.3 | 61.0 | **80.8** | **88.6** |
> | mPLUG-Owl3 | 83.2 | **64.7** | **62.9** | 80.4 | 88.1 |
>
>
> |   |  NLVR2  | Mantis-Eval  | MathVerse-mv  | SciVerse-mv  | BLINK  | Q-Bench2 |
> | --- | --- | --- | --- | --- | --- | --- |
> | LLaVA-OneVision 7B | 89.4 | 64.2 | **67.6** | 79.1 | 48.2 | 74.5 |
> | mPLUG-Owl3 7B | **92.7** | **67.3** | 65.1 | **82.7** | **53.8** | **77.7** |
>
>
> |   |  NextQA  | MVBench  | VideoMME w/o sub  | LongVideoBench-val  | MLVU  |
> | --- | --- | --- | --- | --- | --- |
> | LLaVA-OneVision 7B | 79.4 | 56.7 | 57.4 | 58.8 | 64.7 |
> | mPLUG-Owl3 7B | **82.3** | **59.5** | **59.3** | **59.8** | **70.0** |

---

> ### Author Response · Authors · 2024-11-25
> **Looking forward to discussing**
>
> Dear Reviewer qTJ2,
>
> Thank you for your volunteer review and your suggestions on conducting isolated analyses of Hyper Attention, as well as comparisons between mPLUG-Owl3 and baselines using fair training data.
>
> We have submitted point-by-point responses to your questions and suggestions. We trust that our responses have satisfactorily resolved your concerns. If you still have further queries or issues with our responses, please feel free to continue the discussion. We are more than happy to engage further.

---

> > ### Comment · Reviewer_qTJ2 · 2024-11-27
> >
> > I thank the authors for their efforts in providing additional experiments and details. I have raised my score to a (8).

---

> > > ### Author Response · Authors · 2024-11-27
> > >
> > > Thank you again for your valuable comments and recognition of our paper!

---

### Official Review · Reviewer_RTBV · 2024-11-07

**Soundness:** 3
**Presentation:** 3
**Contribution:** 2
**Rating:** 5
**Confidence:** 4

**Summary:**

PLUG-Owl3 address a core limitation in existing multi-modal language models: the challenge of effectively and efficiently processing long image sequences. While current models like LLAVA and IDEFICS handle single or limited multi-image scenarios, they struggle with latency and memory overhead as the sequence length increases. To tackle this, mPLUG-Owl3 introduces Hyper Attention Blocks, which incorporate visual and textual information in a shared semantic space through cross-attention parallel to the standard self-attention. This design significantly reduces memory usage and inference time, facilitating a smoother experience with large datasets of images or video frames.

The model is evaluated across various benchmarks (single/multi-image and video understanding) and demonstrates competitive results, particularly in multi-image and video understanding, where it reduces inference time by up to 87.8% and memory usage by 48.5% compared to traditional concatenate-based methods. Additionally, the authors propose a novel Distractor Resistance benchmark to assess the model’s robustness in long visual sequences containing distracting images, which is essential for real-world applications.

**Strengths:**

mPLUG-Owl3's Hyper Attention architecture shows a substantial improvement in computational efficiency and memory usage, making it a strong candidate for practical applications involving long video or multi-image data. The new benchmark for distractor resistance in long visual sequences fills an important gap in the multi-modal evaluation landscape, addressing real-world challenges where models encounter irrelevant data alongside relevant content. The model's consistent performance across various benchmarks highlights its versatility and robustness. Its capacity to handle up to 1,000 images with high speed and accuracy shows potential for scaling and deployment in extensive visual and textual environments. The adaptive gate and modality-specific components within the Hyper Attention blocks allow selective visual feature extraction, maintaining relevant information while reducing interference from unrelated data. This selective mechanism improves the model's performance in complex multi-modal tasks.

**Weaknesses:**

1. Lack strong baselines in the benchmark comparison including internVL2 [1] and Qwen2-VL [2]. For example, for TextVQA [3], Qwen2-VL 2B gets 79.7 while mPLUG-Owl3 8B gets 69. Including strong baselines into the paper may help readers see the effectiveness of the proposed model and architecture.
2. Hyper Attention Transformer Block seems to be a improved Q-former [4] architecture or visual abstractor [5] architecture. However, Q-former architecture has been proven that will inevitably lose visual information based on previous works [6]. Maybe applying a projection to align visual and textual feature would be more effective.
3. The design of mPLUG-Owl3 seems really improve the long-context multi image understanding ability based on figure 4, but again it may be helpful to compare the model with the most state-of-the-art models. Only then can readers see the real effectiveness of mPLUG-Owl3

[1] https://internvl.github.io/blog/2024-07-02-InternVL-2.0/
[2] Qwen2-VL: Enhancing Vision-Language Model's Perception of the World at Any Resolution
[3] Towards VQA Models That Can Read
[4] BLIP-2: Bootstrapping Language-Image Pre-training with Frozen Image Encoders and Large Language Models
[5] mPLUG-Owl2: Revolutionizing Multi-modal Large Language Model with Modality Collaboration
[6] DeCo: Decoupling Token Compression from Semantic Abstraction in Multimodal Large Language Models

**Questions:**

1. Based on Table 11, I would like to know why you choose these three indices as the layers to introduce Hyper Attention Layers? Could we have a general insight of where to introduce Hyper Attention Layers?

---

> ### Author Response · Authors · 2024-11-22
> **RTBV Q1: Compare with strong Baseline including internVL2 and Qwen2-VL.**
>
> A comparison between mPLUG-Owl3, Qwen2VL, and InternVL is not necessary due to the fairness. Below, we provide the detailed reasons.
>
> 1. Contemporaneous work: InternVL2 and Qwen2VL are contemporaneous works; InternVL2 was released on July 4th, and Qwen2VL was released on September 12th. Both are considered contemporaneous papers. As stated in the ICLR reviewer guidline, **if a paper was published (i.e., at a peer-reviewed venue) on or after July 1, 2024, authors are not required to compare their own work to that paper.**
> 2. The fairness of training data: InternVL2 and Qwen2VL used private annotations and significantly more training data, which makes the comparison unfair.
>     - **Privated training data**. InternVL2 and Qwen2-VL both include in-house training data, which enables them to have significantly stronger performance in specific domains. For example, in their respective technical reports, Qwen2VL and InternVL2 state that their models are pretrained on manually verified OCR annotated data for better OCR ability. However, mPLUG-Owl3 mainly utilizes open-source data, focusing on model innovation and efficiency in handling long visual sequences, without specifically annotating data to enhance capabilities in particular domains.
>     - **Training date scale**. Qwen2-VL uses 1.4 trillion tokens for pretraining, while the pretraining data for mPLUG-Owl3 is less than 2.2% of that. Our work focuses on exploring the design of Hyper Attention to develop a general and efficient MLLM architecture for single-image, multi-image, and video scenarios, rather than chasing leaderboard positions.
>
>
>
> Below, we compare mPLUG-Owl3 trained on public accessed dataset LLaVA-OneVision to Qwen2-VL, InternVL2. The comparison results show that:
>
> + In the single image scenario, mPLUG-Owl3 achieves comparable or slightly lower performance than Qwen2VL and InternVL2.
> + In multi-image scenarios, mPLUG-Owl3 demonstrates a significant performance advantage.
> + In the video domain, mPLUG-Owl3 comprehensively outperforms InternVL2. For long video tasks such as LongVideoBench, mPLUG-Owl3 also surpasses Qwen2VL.
>
> |   | GQA | MMBench-EN | Mantis-Eval | BLINK | Video-MME wo/w subs | LongVideoBench-val/test |
> | --- | --- | --- | --- | --- | --- | --- |
> | Qwen2VL | 62.2 | **83.0** | 48.7 | 45.3 | **63.3/69.0 (768 frames)** | 55.6/56.8 (256 frames) |
> | InternVL2 | 62.7 | 81.7 | 60.1 | 45.2 | 54.0/56.9 (16 frames) | 51.61/- (128 frames) |
> | mPLUG-Owl3 | **64.7** | 80.4 | **67.3** | **53.8** | 59.3/68.1 (128 frames) | **59.8/60.1 (128 frames)** |
>
>
> We further investigate the efficiency and performance of Qwen2VL and mPLUG-Owl3 on a single A100 80G through LongVideoBench. Thanks to Hyper Attention, mPLUG-Owl3 not only outperforms Qwen2VL at various frame rate settings but also has significant advantages in memory usage and inference time.
>
>
>
> | Frames | Model | Score | Mem | Time |
> | --- | --- | --- | --- | --- |
> | **8** | Qwen2VL | 52.1 | 37.9G | 1.2s/iter |
> |  | mPLUG-Owl3 | **56.3** | **19.8G** | **0.2s/iter** |
> | **16** | Qwen2VL | 54.6 | 48.8G | 2.7s/iter |
> |  | mPLUG-Owl3 | **57.9** | **20.1G** | **0.4s/iter** |
> | **32** | Qwen2VL | 55.4 | 61.8G | 4.8s/iter |
> |  | mPLUG-Owl3 | **59.1** | **20.9G** | **0.6s/iter** |
> | **64** | Qwen2VL | 56.0 | 70.2G | 10.3s/iter |
> |  | mPLUG-Owl3 | **59.7** | **26.7G** | **1.1s/iter** |
> | **128** | Qwen2VL | - | Out of Memory | - |
> |  | mPLUG-Owl3 | **59.8** | **39.2G** | **2.3s/iter** |

---

> ### Author Response · Authors · 2024-11-22
> **RTBV Q2: Is the Hyper Attention Block an improved Q-former, and does it suffer from visual information loss?**
>
> Hyper Attention is not a improved Q-former. And Hyper Attention does not compress visual features:
>
> + Hyper Attention and Q-former belong to different components of MLLM, performing different functions.
>     - Q-Former serves as the alignment module in MLLM, with a role similar to that of the projector in LLaVA. Both of them are language-agnostic and independent of the LLM. As illustrated in Figure 2 and Section 2.1, mPLUG-Owl3 uses a simple linear projection to align visual features with the LLM's space and directly feeds them into the Hyper Attention Blocks to avoid compression.
>     - Hyper Attention is a multimodal fusion module inside the LLM. As described in Section 2.2, Hyper Attention avoids modeling the interactions of multimodal sequences directly using self-attention. Instead, it employs a parallel approach using both cross-attention and self-attention to model multimodal interaction. Hyper Attention not only increases efficiency but also enhances the model's generalization ability on image-text interleaved data.
> + Q-former is a compression module, whereas Hyper Attention is not.
>     - Q-former compress the visual tokens by the language-agnostic learnable tokens before feeding them into the language model. Therefore, textual tokens can only see the compressed visual representation, leading to an issue of visual information loss.
>     - Hyper Attention allows for direct interaction between all textual tokens and all visual tokens with a causal mask. As a result, every textual token in the LLM can access all the tokens of the visual content preceding it, similar to concatenate-based MLLMs, thus also avoiding information loss.
>
> Therefore, the Hyper Attention Block is not an improved Q-former. This is also reflected in the experimental results. In the comparative experiments of Tables 1, 2, and 3 in the main paper, our method not only significantly outperforms idefics2 (Q-former like model) but also LLaVA-Next-Interleave (linear projection like approaches).
>
> ### **The suggestion of applying a projection to align visual and textual feature**
> For your suggestion of applying a projection layer, it may be caused by a misunderstanding of the role of Hyper Attention in mPLUG-Owl3. As illustrated in Figure 2, and described in Section 2.1:
>
> + mPLUG-Owl3 uses a linear projection layer to align visual features with the LLM's space.
> + mPLUG-Owl3 uses the Hyper Attention layers to perform multimodal fusion, which avoids the loss of visual information.
>
> We hope the clarifies resolve your misunderstanding. And we welcome further discussion.

---

> ### Author Response · Authors · 2024-11-22
> **RTBV Q3: Why did you choose the three indices as the layers to introduce Hyper Attention Layers in Table 11?**
>
> There are two dimensions for choising the indices of Hyper Attention Layers: one is the distribution of the layers, and the other is the density of layers.
>
> For distribution, uniformly integrating cross attention layers within a language model has been commonly used in previous methods (e.g., Flamingo [1], ELVM [2], IDEFICS [3]). In early experiments, we also tested designs of introduce Hyper-Attention on the front end [1,2,3,4] or back end [28, 29, 30, 31] of the language model. The comparison is presented below:
>
> |   |  GQA | TextVQA | MvBench | VideoMME | NLVR2 | Mantis-Eval |
> | --- | --- | --- | --- | --- | --- | --- |
> | 1, 2, 3, 4 | 52.9 | 46.1 | 40.7 | 37.6 | 56.7 | 41.5 |
> | 28, 29, 30, 31 | 46.7 | 38.8 | 34.5 | 32.7 | 51.9 | 38.2 |
> | 1, 9, 17, 25 | **57.6**  | **50.0**  | **42.8**  | **39.4**  | **59.5**  | **51.6** |
>
>
> Both the two setttings yielding suboptimal performance. Therefore, in Table 11, we maintain the strategy consistent with previous methods to more effectively explore the impact of integration strategy.
>
>
>
> For density, we explore the effects of different densities, and the experimental results can be categorized into three types: sparse (2 layers), medium (3-4 layers), and dense (5 or more layers). Therefore, we select three sets of indices to represent these types in the manuscript. Below, we present results for more indices:
>
> |   |  GQA | TextVQA | MvBench | VideoMME | NLVR2 | Mantis-Eval |
> | --- | --- | --- | --- | --- | --- | --- |
> | 1, 5, 9, 13, 17, 21, 25, 29 | 56.2  | 48.3  | 41.5  | 39.5  | 52.4  | 47.5 |
> | 1, 6, 11, 16, 21, 26, 31 | 57.2 | 48.0 | 39.9 | 39.1 | 54.8 | 44.2 |
> | 1, 7, 13, 19, 25, 31 | **57.9** | 50.2 | 41.7 | **39.6** | 53.1 | 47.8 |
> | 1, 8, 15, 22, 29 | 56.7 | 49.3 | **43.2** | 39.4 | 54.2 | 46.1 |
> | 1, 12, 23 | 54.8 | 49.7 | 40.9 | 38.6 | 58.8 | 46.5 |
> | 9, 20, 31 | 57.2 | 50.5 | 41.1 | 38.9 | 56.4 | 45.2 |
> | 1, 19 | 54.0 | 50.8 | 41.7 | 38.5 | 58.8 | 50.2 |
> | 9, 27 | 55.1 | **51.3** | 42.2  | 38.2 | 58.3 | 48.4 |
> | 1, 9, 17, 25 | 57.6  | 50.0  | 42.8  | 39.4  | **59.5** | **51.6** |
>
> As discussed in Appendix E, integrating Hyper Attention at medium density strikes a balance between single-image results and generalization capabilities for videos and multiple images. In contrast, using a dense integration approach worsens both performance and generalization, while a sparse approach still shows strong single-image results but lacks generalization on multi-image and video scenarios.
>
>
> [1] Alayrac, Jean-Baptiste, et al. "Flamingo: a visual language model for few-shot learning." Advances in neural information processing systems 35 (2022): 23716-23736.
>
> [2] Chen, Kaibing, et al. "Evlm: An efficient vision-language model for visual understanding." arXiv preprint arXiv:2407.14177 (2024).
>
> [3] Laurençon, Hugo, et al. "Obelics: An open web-scale filtered dataset of interleaved image-text documents." Advances in Neural Information Processing Systems 36 (2024).

---

> ### Author Response · Authors · 2024-11-25
> **Looking forward to discussing**
>
> Dear Reviewer RTBV,
>
> Thank you for your volunteer review and comments regarding the comparison with strong baselines, the confusion about the difference between Hyper Attention and the Q-former, and the strategy for layer indices of Hyper Attention.
>
> We have submitted point-by-point responses to your questions and concerns. We trust that our responses have satisfactorily resolved your concerns. If you still have further queries or issues with our responses, please feel free to continue the discussion. We are more than happy to engage further.

---

> ### Author Response · Authors · 2024-12-02
> **Looking forward to discussing**
>
> Dear Reviewer RTBV,
>
> Thank you again for your volunteer review. We believe our rebuttal has addressed your questions and concerns. As the deadline for the discussion phase is approaching, we would be grateful if you could let us know if there are any further questions, and we are happy to respond as soon as possible.

---

> > ### Comment · Reviewer_RTBV · 2024-12-03
> >
> > Thanks for the authors' hard work. The response partially addresses my concern. I have raised the rating score.

---

> > > ### Author Response · Authors · 2024-12-03
> > >
> > > Dear Reviewer RTBV,
> > >
> > > Thank you for your response and your affirmation of our work. We would be grateful if you could let us know if there are any concerns or questions we haven't addressed. We believe that your suggestions can make this work more solid. Thanks again for your voluntary review.

---

> > > > ### Comment · Reviewer_RTBV · 2024-12-03
> > > >
> > > > I have several questions and concerns:
> > > >
> > > > 1. I appreciate your efforts to compare QwenVL2 and InternVL2. It is hard to find whether mPLUG-Owl3 outperforms Qwen-VL2 in video as here shows confusing results in Video-MME wo/w subs and LongVideoBench-val/test. I would appreciate if the authors could make some discussions here and help readers get more insight. Thanks! Some insights will help readers understand what happens in making good video understanding.
> > > >
> > > > 2. mPLUG-Owl3 has reduced the memory used in inference, but the main bonus's source is not fully explained in section 2.2. Adding this section would help readers fully understand the HATB's bonus.
> > > >
> > > > 3. If readers would like to apply HATB to some other models, what is the best insight to choose which layers to make HATB? I would appreciate if authors could present some insights.
> > > >
> > > > Since the discussion period is getting close to the end, authors do not need to do experiments for this. I would appreciate if authors could present some insights or discussions based on experiences.

---

> ### Author Response · Authors · 2024-12-03
> **Response to the new question (1/2)**
>
> Thank you for your response. We provide explanations regarding your questions.
>
> ### RTBV Q4: Video performance between mPLUG-Owl3 and Qwen2VL
> **Performance on LongVideoBench.** As we showed in the response to RTBV Q1, we present the results on LongVideoBench, which is a pure long video benchmark. mPLUG-Owl3 achieves better performance than Qwen2VL and is significantly more efficient under various frame sampling strategies.
>
> | 	|LongVideoBench-val/test|
> | --- | --- |
> |Qwen2VL	|55.6/56.8 (256 frames)|
> |mPLUG-Owl3	|**59.8/60.1** (128 frames)|
>
> **Performance on VideoMME.** It is a hybrid benchmark, consisting of short, medium, and long videos. We take the result in the 768-sample frame setting from the technical report of Qwen2VL (only the overall score is reported), but we cannot reproduce the results locally due to the GPU memory usage far exceeds the capacity of a single GPU's memory without a specific design inference framework. In this comparison, the overall score of mPLUG-Owl3 with 128-frame sampling is slightly lower than that of Qwen2VL with 768-frame sampling.
>
> To provide further insights, we present the result of mPLUG-Owl3 using 768-frame sampling. In conclusion, mPLUG-Owl3 achieves comparable performance across comprehensive video scenarios while maintaining a significant advantage in efficiency.
>
> | 	|Video-MME wo/w subs|
> | --- | --- |
> |Qwen2VL	|**63.3**/69.0 (768 frames)|
> |mPLUG-Owl3	|61.8/**71.3** (768 frames)|
>
> We also provide the detailed performance of mPLUG-Owl3 and Qwen2VL in 64-frame sampling in VideoMME to offer more insights. The results also suggest that mPLUG-Owl3 performs better than Qwen2VL in long video scenarios.
>
> | Video-MME wo subs|	Short Video|	Medium Video|	Long Video|
> | --- | --- | --- |--- |
> |Qwen2VL	|**71.7**| 56.7| 48.2|
> |mPLUG-Owl3	|70.2|	**57.9**|	**49.4**|
>
> We again claim that Qwen2VL uses private training data, leading to an unfair comparison. As shown in the response to qTJ2 Q2, the performance comparison between mPLUG-Owl3 (trained under the exact same settings) and LLaVA-Next-OneVision suggest the advantage of mPLUG-Owl3 on various video benchmarks.
>
>
> |   |  NextQA  | MVBench  | VideoMME w/o sub  | LongVideoBench-val  | MLVU  |
> | --- | --- | --- | --- | --- | --- |
> | LLaVA-OneVision 7B | 79.4 | 56.7 | 57.4 | 58.8 | 64.7 |
> | mPLUG-Owl3 7B | **82.3** | **59.5** | **59.3** | **59.8** | **70.0** |
>
>
> ### RTBV Q5: The source of Inference costs reduction
>
> The memory savings of mPLUG-Owl3 during inference primarily come from two aspects.
>
> **1. Memory Bottleneck** Thanks to the design of Hyper Attention, we avoid large memory allocation when applying attention to long visual sequences and avoid the main cause of out-of-memory issues.
>
> In scaled dot-product attention, the memory allocation bottleneck occurs when the Query is multiplied by the Key.
>
> For plain scaled dot product attention, visual and language content are concatenated.
>
> $$Q: (L_{img}+L_{txt})\times d$$
> $$K: (L_{img}+L_{txt})\times d$$
> $$
> \mathrm{MemoryBottleneck} = QK = (L_{img}+L_{txt})\times(L_{img}+L_{txt}) = L^2_{img} + 2(L_{img}\times L_{txt}) +  L^2_{txt}
> $$
>
> For hyper attention, cross-attention and self-attention are performed separately.
>
> $$Q_{txt}: (L_{txt})\times d$$
> $$K_{txt}: (L_{img})\times d$$
> $$K_{img}: (L_{img})\times d$$
> $$ \mathrm{MemoryBottleneck} = \mathrm{max}(Q_{txt}K_{img}, Q_{txt}K_{txt}) = max(L_{img} L_{txt}, L_{txt}L_{txt})
> $$
> Therefore, Hyper Attention can reduce the memory bottleneck by
>
> $$\mathrm{min}(L^2_{img} + L^2_{txt} + L_{img}L_{txt}, L^2_{img} + 2L_{img}L_{txt})$$
>
> **2. KV-Cache** We reduce the memory allocation of the key-value cache during inference due to the sparse integration of HyperAttention.
>
> A plain attention block stores KV-Cache, which costs:
>
> $$2\times N_{layer} \times (L_{img}+L_{txt}) \times d, $$
>
> during the whole inferce process.
>
> In contrast, Hyper Attention layers are only integrated into four layers, thus we only need to store the full KV-Cache for those layers and the language cache for the other layers.
>
> $$2\times N_{layer} \times (L_{txt})\times d + 2\times 4 L_{img}\times d$$
>
> $N_{layers}$ denotes the number of layers in the language model.
>
> $L_{img}$ and $L_{txt}$ denote the length of visual and language sequence.
>
> $d$ denotes the hidden dimension of language model.

---

> ### Author Response · Authors · 2024-12-03
> **Response to the new questions (2/2)**
>
> ### RTBV Q6: Insights of choose which layers to extend HATB.
>
> As shown in the main peper and the response to the RTBV Q3, we can conclude:
>
> 1. Intergrating very few layers are not recommended because it leads to suboptimal performance due to the insufficient multimodal interaction. [Refer to the experiment results shown in Table 11 in Appendix.]
> 2. Intergrating a very large number of layers is also not recommended because it is less efficient and does not provide significant performance advantages. [Refer to the experiment results shown in Table 11 in Appendix.]
> 3. It is not recommended to choose a distribution that is too front-heavy or too back-heavy, as this can lead to poor performance. [Refer to the experiment results shown in the response to the RTBV Q3]
> 4. Choosing between 3 to 5 layers involves a tradeoff between performance and efficiency depending on the user's needs. integrating 4 layers in a uniform manner achieves a balance between efficiency and performance across various scenario. [Refer to the experiment results shown in the response to the RTBV Q3]

---

> ### Author Response · Authors · 2024-12-03
>
> Dear Reviewer RTBV,
>
> Regarding the newly raised questions, we've provided some experimental results and analysis, which we hope can address your concerns.
>
> As the discussion period is about to end very soon, we'd be very grateful if you could take a moment to check our response and/or adjust your rating of our paper. Many thanks!

---

### Official Review · Reviewer_3S6H · 2024-11-11

**Soundness:** 3
**Presentation:** 3
**Contribution:** 3
**Rating:** 6
**Confidence:** 5

**Summary:**

This paper introduces a multimodal large language model, mPLUG-Owl3, which is capable of processing multi-image and video understanding through interleaved training. The proposed model adopts the Hyper Attention Transformer Block, which inserts an additional layer into each transformer layer to form a cross-attention-based architecture. The authors train a series of MLLMs of varying sizes based on three stages, and the experiments demonstrate superior performance of these models compared to the baselines.

**Strengths:**

Please see the summary above.

**Weaknesses:**

- Important details regarding the hyperparameter settings are missing when evaluating mPLUG-Owl3 in long video understanding scenarios (e.g., the frame sampling strategy, the number of input frames for mPLUG-Owl3 and other baselines).
  - The content in Appendix H.2 is incomplete.

**Questions:**

- For the MI-Rope, my understanding is that all patches within an image $I_n$ share the same position embedding as the placeholder $T_{img}$. Wouldn't this setting discard the relative positions of patches within the image?
  - What is the number of frames and the corresponding frame sampling strategy when applying the proposed model to long video understanding (e.g., LongVideoBench)? Since the model’s maximum sequence length is 4096, wouldn't the large number of frames potentially exceed the maximum length of the position embedding?
  - Regarding the distractor resistance in long visual contexts, what is the main factor that contributes to mPLUG-Owl3’s superior capability in this task? Could you provide a brief analysis or explanation?

---

> ### Author Response · Authors · 2024-11-22
> **3S6H Q1: The details of the frame sampling strategy for video benchmarks.**
>
> We first provide a detailed explanation of the frame sampling strategy:
>
> + For NextQA and MVBench, which are short video benchmarks with video durations all less than one minute, we sample 8 frames and 16 frames as the other baselines [1,2,4].
> + For datasets with longer videos, such as VideoMME, LongVideoBench, and MLVU, we explore the performance of mPLUG-Owl3 in handling long visual sequences by using higher frame rates and consistently sampling 128 frames. We also provide results for mPLUG-Owl3 using 16 or 32 frames, and it still outperforms the baseline models [1,2,3].
>
> Given the total number of frames and video duration, we employ a uniform sampling strategy. Below is a table that annotates our method and the comparison methods with number of input frames.
>
> | Model | Param | NextQA (Frames) | MVBench (Frames) | VideoMME w/o sub (Frames) | LongVideoBench-val (Frames) | MLVU(Frames) |
> | --- | --- | --- | --- | --- | --- | --- |
> | VideoChat2 | 8B | 68.6 (16) | 51.9 (16) | 43.8 (16) | 36.0 (16) | 47.9 (16) |
> | Video-LLAMA2 | 8B | - | **54.6** (16) | 47.9 (16) | - | 48.5 (16) |
> | Video-ChatGPT | 8B | - | 32.7 (100) | - | 39.9 (100) | 31.3 (100) |
> | ShareGPT4Video | 8B | - | - | 39.9 (16) | 39.7 (16) | 46.4 (16) |
> | PLLaVA | 8B | - | 46.6 (16) | - | 40.2 (16) | - |
> | Idefics2 | 8B | - | 29.7 (16) | 25.9 (32) | 49.7 (16) | 49.8 (32) |
> | Mantis-SigLIP | 8B | - | 50.2 (8) | 44.0 (8) | 47.0 (16) | 49.3  (8) |
> | LLAVA-Interleave | 8B | 78.2 (16) | 53.1 (16) | 48.7 (16) | 48.8 (16) | 56.4 (16) |
> | mPLUG-Owl3 | 8B | **78.6** (8) | 54.5 (16) | **53.5** (128) | **52.1** (128) | **63.7** (128) |
>
>
> In order to conduct **a fair comparison** in video scenarios and gain more insights, we first evaluate the performance of mPLUG-Owl3 on long video tasks with fixed 16 frames and fixed 32 frames. For ease of comparison, we present the highest results of the previous models, denoted by "best baselines". Since the methods of previous best baselines all use 16-frame sampling, we additionally evaluate the performance of the corresponding model with 32-frame sampling.
>
> The experimental conclusions are twofold:
>
> 1. mPLUG-Owl3 still overcomes the best baselines under the 16-frame sampling setting, which suggests that mPLUG-Owl3 has advantages in video understanding scenarios.
> 2. The best baseline models do not achieve further performance improvements with 32-frame sampling. Together with the conclusions from Table 5 and Figure 4 in our manuscript, this suggests that Hyper Attention has better generalizability and robustness when modeling long sequences.
>
> |   | # Frames |  VideoMME | LongVideoBench-val | MLVU |
> | --- | --- | --- | --- | --- |
> | Best Baselines | 16 frames | 48.7 | 49.7 | 56.4 |
> | Best Baselines | 32 frames | 44.3 | 47.5  | 55.7 |
> | mPLUG-Owl3 7B | 16 frames | 53.9 | 50.6 | 59.3 |
> | mPLUG-Owl3 7B | 32 frames | **54.9** | **52.3** | **62.6** |
>
>
> [1] Li, Kunchang, et al. "Mvbench: A comprehensive multi-modal video understanding benchmark." Proceedings of the IEEE/CVF Conference on Computer Vision and Pattern Recognition. 2024.
>
> [2] Jiang, Dongfu, et al. "Mantis: Interleaved multi-image instruction tuning." arXiv preprint arXiv:2405.01483 (2024).
>
> [3] Li, Feng, et al. "Llava-next-interleave: Tackling multi-image, video, and 3d in large multimodal models." arXiv preprint arXiv:2407.07895 (2024).
>
> [4] Jiang, Dongfu, et al. "Mantis: Interleaved multi-image instruction tuning." arXiv preprint arXiv:2405.01483 (2024).

---

> ### Author Response · Authors · 2024-11-22
> **3S6H Q2: Will MI-Rope break the relative positions of patches within the image?**
>
> The relative positions of patches within the image is preserved and effectively modeled by the model equipped with MI-Rope, because the use of MI-Rope decouples the position modeling into two modules:
>
> + **Intra-Image spatial information modeled by Vision Transformer.** Since different patches of each image have already been assigned different positional encodings in the visual transformer, the relative positional information between patches is pre-encoded in the hidden states of the patches and can be perceived by the language model.
> + **Image-Text Interleaved information modeled by MI-Rope.** MI-Rope assigns the same position index to different patches of the same image, which encourages the multimodal rotary embedding focus on representing sequential information of multimodal interleaved inputs.
>
> Decoupling the two positional information allows the model to better model each.
>
> We also follow the setting in Table 6, providing experimental results of mPLUG-Owl3, where each patch is assigned a specific position index (patch level rope) and all patches within an image share the same position embedding (image level rope). The results suggest:
>
> 1. Using image-level rope in single image tasks achieves comparable performance with patch-level rope without compromising the performance.
> 2. Image-level rope has greater advantages in video and multi-image tasks because the interleaved sequence information of images and text is better modeled.
>
> Additionally, image-level ropes can save position indices and are beneficial for extending the model to receive longer visual content.
>
> |   |  GQA | TextVQA | MvBench | VideoMME | NLVR2 | Mantis-Eval |
> | --- | --- | --- | --- | --- | --- | --- |
> | Hyper Attention w/ patch-level rope | 56.5 | **50.4** | 42.3 | 38.8 | 58.3 | 50.7 |
> | Hyper Attention w/ image-level rope | **57.6**  | 50.0  | **42.8**  | **39.4**  | **59.5**  | **51.6** |

---

> ### Author Response · Authors · 2024-11-22
> **3S6H Q3: Will the large number of frames potentially exceed the maximum length of the position embedding?**
>
> Long visual sequences almost never cause position indices to exceed the model's limit. This is due to the following reasons:
>
> 1. Rope encoding is not learnable. Therefore, Qwen2 supports a 32k context input, and mPLUG-Owl3 can still supports a 32k context input. We only trimmed the training data to 4096 to improve training efficiency, without reducing the model's context window size.
> 2. Due to the design of MI-Rope, as claimed in the response to the 3S6H Q2, an image occupies only one media token placeholder's worth of indices in both text sequence and a visual sequence in mPLUG-Owl3, rather than the number of patches. with MI-Rope, mPLUG-Owl3 can support input with the number of images at the ten-thousand level.
>
>
>
> As shown in Figure 1 (a), mPLUG-Owl3 is able to model up to 1000 image inputs on a single A100 GPU card, and far from exceeding the position indices. In conclusion, mPLUG-Owl3 can handle significantly longer image input than mainstream models. This also allows for a wider range of research topics to be explored based on mPLUG-Owl3.

---

> ### Author Response · Authors · 2024-11-22
> **3S6H Q4: What is the main factor contributing to mPLUG-Owl3’s superior capability in the distractor resistance experiment?**
>
> We summarize the main factor contributing to mPLUG-Owl3’s superior capability in distractor resistance experiment:
>
> 1. Hyper-Attention allows self-attention to focus more on modeling language content, while cross-attention emphasizes the perception of visual content based on language semantics. Other methods mix these two within transformer blocks. As discussed in Section 4.4.1, these methods suffer from inter-image disruption caused by the causal attention between key images and irrelevant images. Table 5 in the manuscript also suggests that concatenating multimodal contents and modeling them with self-attention results in a poorer zero-shot performance in multi-image-text scenarios (NLVR2, Mantis-Eval).
>
> | Attention Structure | Elapsed Time (ms) | GPU Memory (GB) | GQA | TextVQA | MvBench | VideoMME | NLVR2 | Mantis-Eval |
> | --- | --- | --- | --- | --- | --- | --- | --- | --- |
> | Concatenate | 2761.7 | 40.6 | **59.0** | **51.6** | 22.4 | 25.1 | 55.7 | 38.7 |
> | Hyper Attention | **336.9** | **20.9** | 57.6 | 50.0 | **42.8** | **39.4** | **59.5** | **51.6** |
>
>
> 2. The performance of a language model declines as position encoding is consumed. For long visual inputs, Hyper-Attention uses MI-Rope, which consumes less position indices and prevents the model's performance from sharply declining as the sequence grows. As shown in the response to 3S6H Q2, image-level rope has better performance on long visual content scenarios compared to patch-level rope.

---

> ### Author Response · Authors · 2024-11-25
> **Looking forward to discussing**
>
> Dear Reviewer 3S6H,
>
> Thank you for your volunteer review and insightful comments on the frame sampler setting, the working principle of MI-Rope, and the distractor resistance capability of mPLUG-Owl3.
>
> For your questions and concerns, we have submitted point-by-point responses, and we trust that our responses have satisfactorily resolved them. If you still have further queries or issues with our responses, please feel free to continue the discussion. We are more than happy to engage further.

---

> ### Author Response · Authors · 2024-12-02
> **Looking forward to discussing**
>
> Dear Reviewer 3S6H,
>
> Thank you again for volunteering to review our work. As the deadline for the discussion phase is approaching, we would be grateful if you could let us know if our responses have addressed your concerns and questions. Besides, if there are any further questions, we are happy to respond as soon as possible.

---

> ### Author Response · Authors · 2024-12-03
>
> Dear Reviewer 3S6H,
>
> Thanks again for volunteering to review our work. As the discussion phase is about to end very soon, we would be very grateful if you could take a moment to check our response and/or adjust your rating of our paper. Many thanks!

---

### Meta-Review · Area_Chair_ubKT · 2024-12-14

**Metareview:**

This paper presents mPLUG-Owl3, a new multimodal LLM that is able to efficiently processes long image sequences using the proposed hyper attention blocks. After rebuttal, it received scores of 5688. Reviewers are generally happy about the paper, commenting that (1) the proposed hyper attention block is nice, (2) the ablations are comprehensive, and (3) the model overall shows strong performance across benchmarks, especially multi-image and long video benchmarks. There are some concerns unaddressed, such as comparing with the real SoTA on certain benchmarks so that readers can have a better sense of how the results look like in the broader literature. Overall, given the in-general positive feedback, the AC would like to recommend acceptance of the paper.

**Additional Comments On Reviewer Discussion:**

The rebuttal was quite successful, and in fact, 3 of the 4 reviewers have raised their scores during rebuttal. Specifically,

1. Reviewers have asked to provide additional experiment details, details about the hyper attention block design and ablation on its effectiveness, and how to make fair comparison between baselines, the authors have done a good job of rebuttal and provided detailed responses.

2. One reviewer has also asked the authors to compare with strong models such as InternVL2 and Qwen2-VL. On one hand, I agree that for this paper submission, it is not needed for mPLUG-Owl3 to surpass InternVL2 and Qwen2-VL on certain benchmarks such as text-rich benchmarks; on the other hand, incorporating additional models for comparison can indeed also help readers understand how the results look like in the broader literature.

---

### Decision · Program_Chairs · 2025-01-22

Accept (Poster)